

**CryoSat Ice Baseline-D Validation and Evolutions**
**Marco Meloni[a],** Jerome Bouffard[b], Tommaso Parrinello[b], Geoffrey Dawson[c], Florent Garnier[d],
Veit Helm[f], Alessandro Di Bella[e], Stefan Hendricks[f], Robert Ricker[f], Erica Webb[g], Ben
Wright[g], Karina Nielsen[e], Sanggyun Lee[i], Marcello Passaro[j], Michele Scagliola[h], Sebastian
Bjerregaard Simonsen[e], Louise Sandberg Sørensen[e], David Brockley[i], Steven Baker[i], Sara
Fleury[d], Jonathan Bamber[c], Luca Maestri[h], Henriette Skourup[e], Rene' Forsberg[e], LorettaMizzi[k]
a.  Serco c/o ESA, Earth Observation Directorate, Via Galileo Galilei, 2 –00044 Frascati, Italy;
b.  ESA (European Space Agency), Earth Observation Directorate, Via Galileo Galilei, 2 –
00044 Frascati, Italy;
c.  Bristol Glaciology Centre, School of Geographical Sciences, University of Bristol, Bristol,
UK;
d.  LEGOS, University of Toulouse, CNRS, IRD, CNES, UPS, (Toulouse), France;
e.  DTU Space, National Space Institute, Department of Geodynamics, Technical University
of Denmark, Kongens Lyngby, Denmark;
f.  Alfred Wegener Institute, Helmholtz Centre for Polar and Marine Research
Klussmanstr. 3D, 27570 Bremerhaven, Germany;
g.  Telespazio VEGA UK Ltd., 350 Capability Green, Luton, Bedfordshire LU1 3LU, UK;
h.  Aresys S.r.l., via Privata Flumendosa, 16, 20132, Milan, Italy;
i.  University College London at The Mullard Space Science Laboratory, Holmbury St Mary,
RH5 6NT, UK;
j.  Deutsches Geodatisches Forschungsinstitut (DGFI/TUM), Munchen, Germany;
k.  Telespazio, Via Tiburtina, 965, 00156 Rome, Italy.
**Corresponding Author:** Meloni, M., marco.meloni@esa.int




**Abstract**
The ESA Earth Explorer CryoSat-2 was launched on 8 April 2010 to monitor the precise
changes in the thickness of terrestrial ice sheets and marine floating ice. For that, CryoSat orbits
the planet at an altitude of around 720 km with a retrograde orbit inclination of 92 ° and a
''quasi'' repeat cycle of 369 days (30 days sub-cycle). To reach the mission goals, the CryoSat
products have to meet the highest quality standards to date, achieved through continual
improvements of the operational processing chains. The new CryoSat Ice Baseline-D, in
operation since 27th May 2019, represents a major processor upgrade with respect to the
previous Ice Baseline-C. Over land ice the new Baseline-D provides better results with respect
to previous baseline when comparing the data to a reference elevation model over the
Austfonna ice cap region, improving the ascending and descending crossover statistics from
1.9 m to 0.1 m. The improved processing of the star tracker measurements implemented in
Baseline-D has led to a reduction of the standard deviation of the point-to-point comparison
with the previous star tracker processing method implemented in Baseline-C from 3.8 m to 3.7
m. Over sea ice, the Baseline-D improves the quality of the retrieved heights in areas up to ~12
km inside the Synthetic Aperture Radar Interferometric (SARIn or SIN) acquisition mask,
which is beneficial not only for freeboard retrieval, but for any application that exploits the
phase information from SARIn Level-1 (L1) products. In addition, scatter comparisons with
the Beaufort Gyre Exploration Project (BGEP, https://www.whoi.edu/beaufortgyre) and
Operation IceBridge (OIB, Kurtz et al., 2013) in-situ measurements confirm the improvements
in the Baseline-D freeboard product quality. Relative to OIB, the Baseline-D freeboard mean
bias is reduced by about 8 cm, which roughly corresponds to a 60% decrease with respect to
Baseline-C. The BGEP data indicate a similar tendency with a mean draft bias lowered from
0.85 m to -0.14 m. For the two in-situ datasets, the Root Mean Square Deviation (RMSD) is
also well reduced from 14 cm to 11 cm for OIB and with a factor 2 for BGEP. Observations



over inland waters, show a slight increase in the percentage of "good observations" in Baseline-
D, generally around 5-10 % for most lakes. This paper provides an overview of the new Level-
1 and Level-2 (L2) CryoSat ice Baseline-D evolutions and related data quality assessment,
based on results obtained from analysing the 6-month Baseline-D test dataset released to
CryoSat expert users prior the final transfer to operations.

**Keywords:** CryoSat; Altimetry; Cryosphere; Ice product status; Instrument performance;
Long-term stability; Ice product evolutions












## 1 Introduction


To better understand how climate change is affecting the Polar Regions in terms of diminishing
ice cover as a consequence of global warming, it remains an urgent need to determine more
precisely how the thickness of the ice is changing, both on land and floating on the sea, as also
detailed in the last IPCC special report on Ocean and Cryosphere
(https://www.ipcc.ch/srocc/download-report/).
In this respect, the ESA Earth Explorer CryoSat-2 (hereafter CryoSat), monitors the changes
in the thickness of marine ice floating in the polar oceans and of the variations in the thickness
of vast ice sheets which are contributing to global sea level rise. To achieve its primary mission
objectives, the CryoSat altimeter is characterised by three operating modes, which are activated
according to a geographic mode mask: 1) pulse width limited Low Resolution Mode (LRM),
2) pulse width limited and phase coherent single channel Synthetic Aperture Radar (SAR)
mode and 3) the dual channel pulse width and phase coherent Synthetic Aperture Radar
interferometric (SARIn) mode.
The CryoSat data are operationally processed by ESA over both ice and ocean surfaces using
two independent processors (ice and ocean), generating a range of operational products with
specific latencies. The ice processor generates Level 1B (L1B) and Level 2 (L2) offline
products typically 30 days after data acquisition for the three instrument modes: LRM, SAR
and SARIn. The ice products are currently generated with the Ice Baseline-D processors. In
addition, Near Real Time (NRT) products are also generated with a latency of 2-3 hours after
sensing to support forecasting services. Details on the previous historic CryoSat ice processing
chain and main L1B and L2 processing steps are reported in Bouffard et al., 2018b. CryoSat
ocean products are instead generated with the Baseline-C CryoSat Ocean Processor (more
details in Bouffard et al., 2018a). An overview of the current CryoSat data products is reported
in Figure 1.





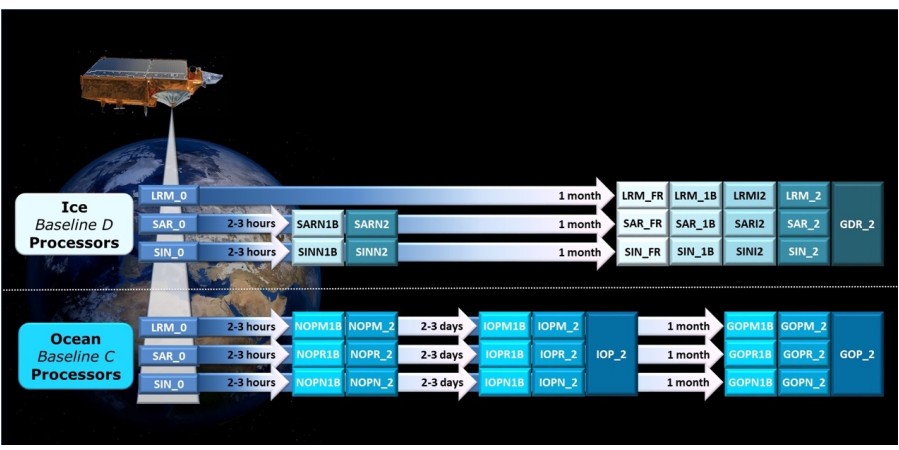


**Figure 1 CryoSat Data Products overview. Map Data ©2019 Google**

In order to achieve the highest quality of data products, and meet mission requirements, the
CryoSat Ice and Ocean processing chains are periodically updated. Processing algorithms and
associated product content are regularly improved based on recommendations from the
scientific community, Expert Support Laboratories, Quality Control Centres and validation
campaigns. In this regard, the new CryoSat Ice Baseline-D processors have been developed
and tested. An Ice Baseline-D Test Data Set (TDS) covering three different time periods
(September - November 2013, February - April 2014 and April 2016 (only SARIn)) was made
available to the CryoSat Quality Working Group (QWG) and scientific experts in order to
opportunely validate and quality check the new products. This paper provides an overview of
the CryoSat Ice Baseline-D evolutions of the processing algorithms and focuses on the in-depth
validation performed on the TDS over land ice, sea ice and inland waters domains. The transfer
to operations of the new CryoSat Ice Baseline-D processors was performed on 27[th] May 2019
and a complete mission data reprocessing is on-going in order to provide users with
homogeneous and coherent CryoSat ice products for proper data exploitation and analysis.
The paper is structured as follows. Section 2 provides an extensive analysis of the major
evolutions included in the Baseline-D separated between L1B and L2 processing stages,

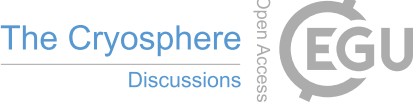

describing the improvements that have been implemented and included in the new baseline
version. Section 3 describes, based on the analysis of the 6-month TDS provided by ESA, the
main validation results in different domains such as land ice, sea ice and inland waters. Section
4 reports the conclusions.





## 2    CryoSat Ice Baseline-D Evolutions


The new Ice Baseline-D processors were approved and transferred to operation on 27th May
2019. A complete list of the evolutions and changes implemented in Baseline-D can be found
in the technical note available at https://earth.esa.int/documents/10174/125272/CryoSat-
Baseline-D-Evolutions while a concise overview of the CryoSat L1B and L2 ice products is
available at https://earth.esa.int/documents/10174/125272/CryoSat-Baseline-D-Product-
Handbook. This revision of the document has been released to accompany the delivery of
Baseline-D CryoSat products. Details about CryoSat and main changes are described below
separated between the L1B and L2 processing stages.

### 2.1    Ice Baseline-D L1B Evolutions


Prior to Baseline-D, the Ice Baseline-C processors were installed on the operational and
reprocessing platforms and Baseline-C L1B products were produced and distributed to users
since the 1st of April 2015 (Scagliola and Fornari, 2015). During this period some issues were
identified and the scientific community suggested a series of evolutions that have been taken
into consideration when updating the L1B processors at Baseline-D. L1B products are now
generated using the new Baseline-D L1B processors, in which software issues have been fixed
and new processing algorithms have been implemented  (for more details refer to the Baseline-
D    products    evolutions    document    available    at
https://earth.esa.int/documents/10174/125272/CryoSat-Baseline-D-Evolutions). One of the
main quality improvements implemented at Baseline-D is the migration from Earth Explorer
Format (EEF) to Network Common Data Form (NetCDF). In addition, it has been shown that
the phase information available in the CryoSat SARIn acquisition mode can be used to reduce
the uncertainty affecting sea ice freeboard retrievals (Armitage et al., 2014). The previous
Baseline-C has shown large negative freeboard estimates at the boundary of the SARIn



acquisition mask, caused by a bad phase difference calibration. In Baseline-D the accuracy of
the phase difference has been improved as well as the quality of the freeboard at the SARIn
boundaries, reducing drastically the percentage of negative retrievals from 25.8% to 0.8% (Di
Bella et al., 2019). In SAR altimetry processing, after the beam forming process, stacks are
formed. A stack is the collection of all the beams that have illuminated the same Doppler cell
(as described in http://www.altimetry.info/filestorage/Radar_Altimetry_Tutorial.pdf). At
Baseline-D, two additional stack characterisation parameters (also known as Beam Behaviour
Parameters) have been added to the SAR/SARIn L1B products. The stack peakiness (Passaro
et al., 2018) can be useful to improve the sea ice discrimination, and the position of the centre
of the Gaussian that fits the range integrated power of the single look echoes within a stack as
function of the look angle (Scagliola et al., 2015). In radar altimetry, the window delay refers
to the 2-way time between the pulse emission and the reference point at the centre of the range
window. The window delay in Baseline-D L1B products now compensates for the Ultra Stable
Oscillator (USO) correction, which is the deviation of the frequency clock of the USO from
the nominal frequency. The L1B users no longer need to apply this correction. In addition, the
accuracy of the mispointing angles has increased by properly considering in their computation
the so called "aberration correction" (more details in Scagliola et al., 2018).

**2.2    Ice Baseline-D L2 Evolutions**
The Baseline-D update to the CryoSat L2 processing fixes a number of anomalies and
introduces    several    processing    algorithm    improvements,    as    described    in
https://earth.esa.int/documents/10174/125272/CryoSat-Baseline-D-Evolutions. In addition to
corrections and improvements, the L2 products are now generated in netCDF format and
contain all previous parameters as well as some new ones. For example, in previous baselines,
the freeboard sea ice processing was restricted to SAR mode regions, resulting in large gaps in



coverage around the coast and in other regions of the Arctic region operating in SARIn. In
Baseline D, the sea ice parameters are also computed over these regions. The height value is
still that from the SARIn mode specific retracking, but new fields have been added to contain
the sea ice processing height result, and freeboard and sea level anomalies are now computed
in SARIn mode (previously SAR mode only). In addition, a new retracker is used for retracking
diffuse waveforms from sea ice regions, and for all waveforms in non-polar regions (more
details in the CryoSat Design Summary Document available at
https://earth.esa.int/documents/10174/125272/CryoSat-L2-Design-Summary-Document).
Over sea ice, the discrimination algorithm used to determine if individual records represent sea
ice floes, leads in the sea ice, or ice-free ocean has been improved with the implementation of
a new discrimination metric based on the peakiness of the stack of SAR beam waveforms. This
method, improves the capability of the algorithm to reject waveforms contaminated by off-
nadir specular reflections (as described in
https://earth.esa.int/documents/10174/125272/CryoSat-L2-Design-Summary-Document).
Some tuning of the thresholds for the other metrics has also been performed, based on analysis
of the test datasets. For the land ice domain, new slope models have been generated, using the
Digital Elevation Models (DEMs) of Antarctica and Greenland described in Helm et al. (2014).
These models were created with more recently acquired data and therefore better represent the
slope of the surface during the period of the CryoSat mission. The DEMs were sampled at high
resolution to derive the surface slope to make the correction more responsive to changes in
slope. Lastly, several improvements have been made to the contents of the L2 products. The
surface type mask model used to discriminate different types of targets, has been updated (as
described in the Baseline-D product handbook available at
https://earth.esa.int/documents/10174/125272/CryoSat-Baseline-D-Product-Handbook).
Variables have been added to the netCDF to explicitly cross-reference the 1 Hz and 20 Hz data.



Finally, the retracker-corrected range to the surface has been added to the product (in addition
to the height).



### 3 CryoSat Ice Baseline-D Validation of Test Dataset Results

#### 3.1 Data Quality: Ice Baseline-D Test Data Verification by IDEAS+

All CryoSat data products are routinely monitored for quality control by the ESA/ESRIN Sensor Performance, Products and Algorithms (SPPA) office with the support of the Instrument Data quality Evaluation and Analysis Service (IDEAS+). In preparation for the Ice Baseline-D, IDEAS+ performed Quality Control (QC) checks on test data generated with the new Ice Baseline-D processors (IPF1 vN1.0 & IPF2 vN1.0). For testing and validation purposes a 6-month TDS was generated at ESA on a dedicated processing environment for two periods: September – November 2013; February – April 2014. IDEAS+ performed QC of a 10-day sample of L1B and L2 data, to assess data quality and check for major anomalies. Following this QC checks, this 6-month TDS was made available to the CryoSat QWG for more detailed scientific analysis.

The content of the product header files (.HDR) was checked to confirm that all Data Set Descriptors (DSDs) were present and correct and all header fields were correctly filled. Similarly, the global attributes section of the netCDF has been checked to ensure data files was consistent and complete. The test Baseline-D products were also checked for any unexpected flags, that may indicate processing anomalies, and all external geophysical corrections were checked to ensure that they were computed correctly. Some minor unexpected changes to the configuration of particular flags was observed as well as the incorrect scaling of the altimeter wind speed values. These minor issues have been resolved in the final Baseline-D release, which has been implemented into operations.





### 3.2 Land Ice

#### 3.2.1 Impact of algorithm evolution on land ice products

CryoSat L1B and L2 products generated using the Baseline-C are the primary input to obtain elevation change time series of the large ice sheets. As those time series are the primary data set to obtain ice sheet wide mass balance and therefore the contribution to sea level change, a consistent high quality CryoSat L1B/2 product is essential. To derive mass balance estimates the Alfred Wegener Institute (AWI) processing chain was used, introduced by Helm et. al. 2014, including TFMRA (Threshold First-Maximum Retracker Algorithm) re-tracking and the refined slope correction (Roemer, et. al., 2007) for LRM mode as well as an interferometric processing using phase and coherence for the SARIn mode L1B data products. In addition, several other groups rely on high quality L1B and L2 data products to generate time series of elevation, respectively mass change (e.g. Nilsson et al., 2015; Simonsen et al, 2017; McMillan et al., 2014; Schroeder et al, 2019). Next to the conventional along track processing, the swath mode has been developed and explored by several groups (Gray et al., 2013; Gourmelen et al., 2017). It has been demonstrated that swath products can be used to estimate basal melt rates of ice shelves (Gourmelen et al., 2017) or high-resolution elevation change time series within the steep margins of the Greenland ice sheet or Arctic Ice Caps. However, a small attitude angle error interpreted as a mispointing error has been observed, which is critical for the accuracy of swath mode products. Bouffard et al., 2018b presented an attitude correction to be applied to Baseline-C products, which should help to reduce this uncertainty. To estimate the impact of the algorithm evolution of the CryoSat Ice Processor to Baseline-D on land ice data records, L2 type products for Baseline-C and Baseline-D were compute using the AWI processing chain. In addition, Level 2 "In-depth" (L2I) product retracker and slope corrections were implemented in the individual data sets to be compared. In a first instance single tracks crossing the Antarctic ice sheet were compared on a point to point basis for all of the individual



parameters included in the L1B and L2I products. Most of the parameters were found to show
close agreement, however a constant offset was found for sigma0 for all of the implemented
LRM L2 retrackers: 0.6 dB, 0.63 dB, 0.65 dB for Ocean, Ice1, Ice2 retracker respectively. This
needs to be considered, as long as both Baselines are used in combination to estimate elevation
change time series, as some groups incorporate a sigma0 correlated correction. Furthermore,
Baseline-D uses an updated surface type mask. This has significantly improved in the ice shelf
area around Antarctica, as shown in Figure 2 for the Filchner-Ronne ice shelf.

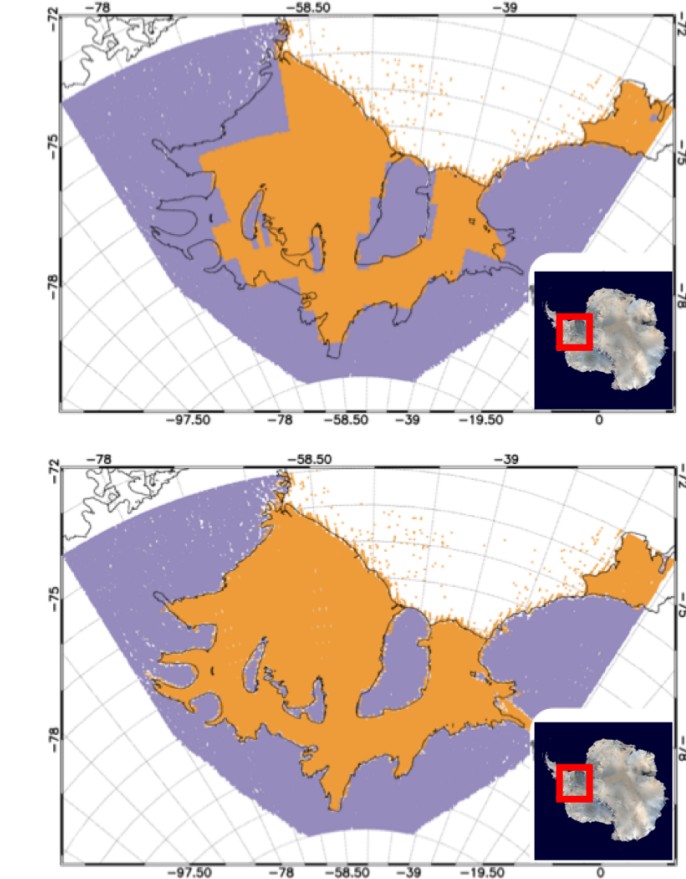


**Figure 2 Surface Type mask shown for the Filchner Ronne ice shelve area. Upper panel Baseline-C; Lower panel**

**Baseline-D. Map Data ©NASA/Dave Pape**





Now, this mask can be applied to differentiate between floating and grounded ice. In addition,
a new slope model for Antarctica, which is based on the elevation model of Helm et al., 2014,
is implemented in Baseline-D. This slightly changes the LRM slope corrected elevation as is
demonstrated for a large area in East Antarctica in Figure 3.

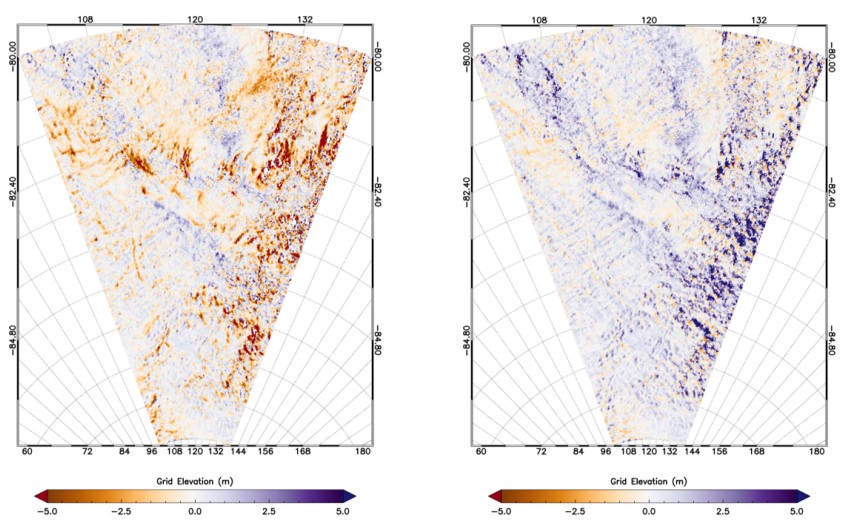

**Figure 3 Differences of slope corrected LRM data to reference DEM (REMA) in East Antarctica.**

**Left: Baseline-C: +0.13 +/- 1.2m, right Baseline-D: -0.11 +/- 1.11 m**

Differences to an independent Antarctic elevation model (REMA) are shown for both
Baselines. The differences vary spatially and the overall mean changed from +0.13 m to -0.11
m. This needs to be considered when estimating time series using data both Baselines, until the
full mission reprocessing is finished. The attitude information for SARIn, such as Roll, Pitch
and Yaw were updated for Baseline-D, incorporating the correction found by Bouffard at al.,
2018b. The correction is as expected and agrees with the auxiliary product already delivered
by ESA. This has negligible effect for SARIn Point Of Closest Approach (POCA) elevations,
however major improvements for swath processed data as shown in Figure 4 and Figure 5.
Figure 4 subpanels show the difference of swath processed data for ascending and descending





tracks to a reference elevation model derived from TanDEM-X data from 2012 for the
Austfonna icecap, respectively. The large positive anomaly is a known glacier surge event
(McMillan et al., 2014). The negative anomaly observed by descending tracks in the eastern
part and the discrepancy between ascending and descending tracks in the western part in
Baseline-C (subpanel B) could be reduced. More clearly, Figure 5 shows this improvement in
the crossover statistics. With the upcoming Baseline-D a correction term as suggested by Gray
et al., 2017, is not needed any more and might not be appropriate as a static correction to
Baseline-C, as the angle correction is variable in space and time.

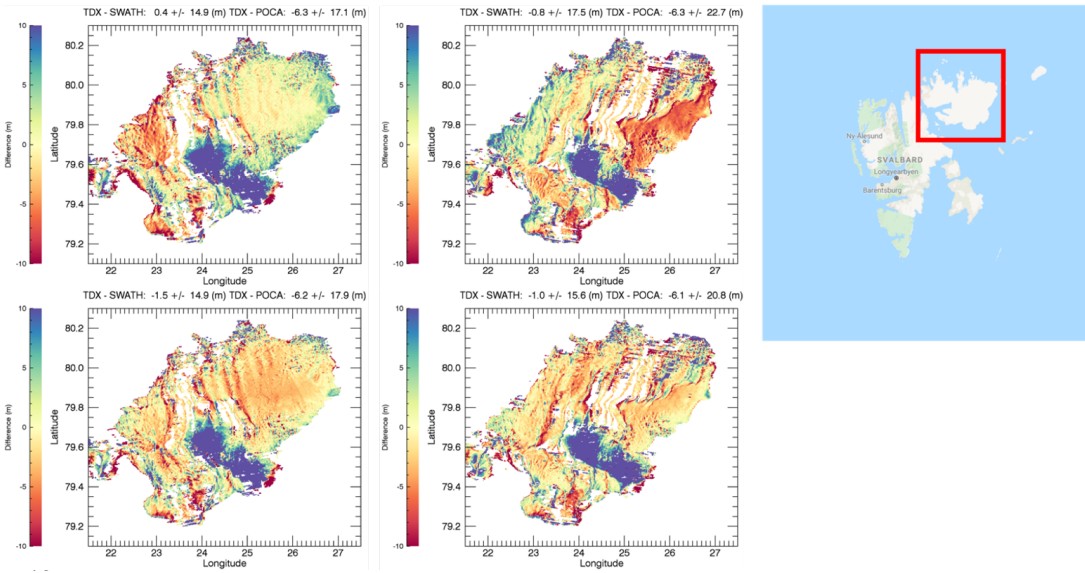

**Figure 4 Differences to reference elevation model derived from TanDEM-X data from 2012 across the Austfonna ice cap. Upper left: ascending Baseline-C, Upper right: descending Baseline-C, Lower left: ascending Baseline-D, Lower right: descending Baseline-D. Map Data ©2019 Google**





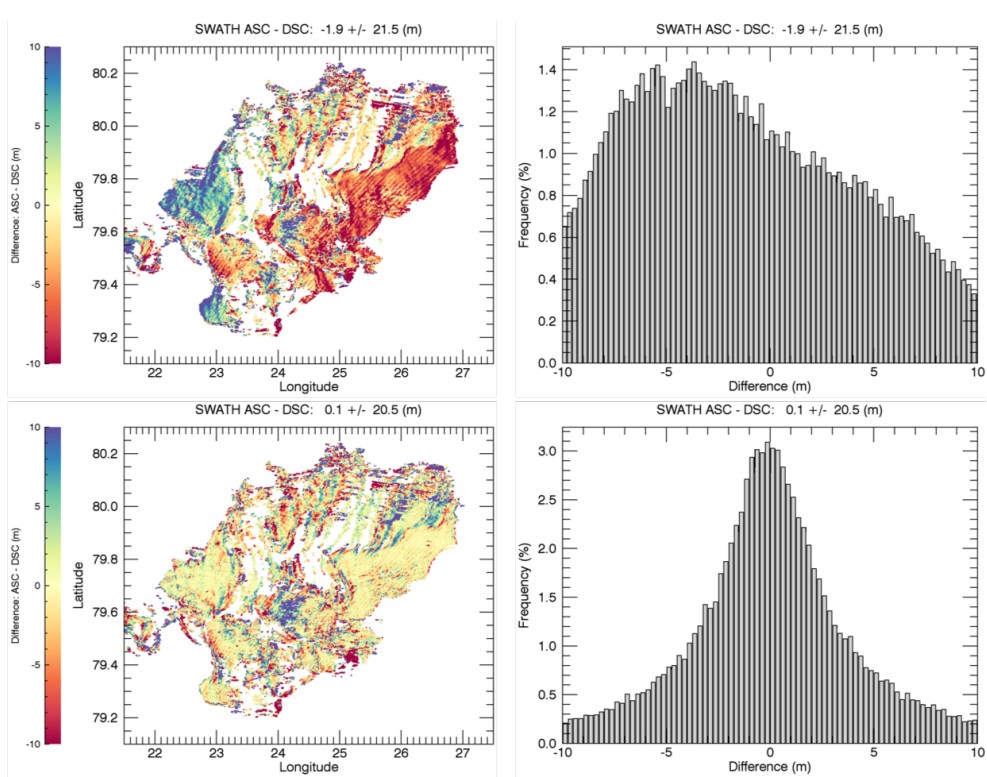

**Figure 5 Crossovers between ascending and descending swath data across Austfonna ice cap. Upper panels: Baseline-C + statistics. Lower panels: Baseline-D + statistics**

### 3.2.2 Baseline-D SARIn swath data over Antarctica

Standard radar altimetry relies on the determination of the Point Of Closest Approach (POCA), sampling a single elevation beneath the satellite. Using CryoSat's interferometric mode (SARIn), it is possible to resolve more than just the elevation at the POCA. If the ground terrain slope is only a few degrees, the CryoSat altimeter operates in a manner such that the interferometric phase of the altimeter echoes may be unwrapped to produce a wide swath of elevation measurements across the satellite ground track beyond the POCA. Swath processing also provides a near continuous elevation field, making it possible to form digital elevation models and to map rates of surface elevation change at a true resolution of 500 m, an order of magnitude finer than is the current state of the art for the continental ice sheets (Gourmelen et





al., 2018). To assess the performance of swath data derived from Baseline-C and Baseline-D
CryoSat L1B data, a point-to-point comparison was performed over the Siple Dome,
Antarctica. This comparison gave a measure of the precision of swath elevation measurement
and allowed for a comparison of each Baseline. The Siple Dome region has been chosen as it
is a relatively stable area with large areas of constant sloping terrain, ensuring a high sampling
density of swath data.
The Baseline-D TDS from February – April 2014 and the Baseline-C data from the same time
period were used in this assessment. Baseline-C data were used with both the original star
tracker measurements and with revised measurements provided by ESA. These were supplied
as a result of an incorrect mispointing angle for the aberration of light being implemented in
Baseline-C, which led to an error in the calculation of the roll of the satellite. Any error in the
roll will result in an error in the geolocation and derived height, and this was shown to decrease
the performance of swath measurements (Gray et al., 2017). Swath data were processed
following Gray et al., 2013, with a minimum coherence and power threshold of 0.9 and -180
dB respectively. For the point-to-point comparison, the closest individual swath elevation
measurement from a different satellite pass was used. A comparison was only made if the
maximum distance between the two geolocated elevation measurement was below 30 m.
Overall 157,000 points were compared at an average distance of 19 m. As the points compared
were distributed over sloping terrain, any difference in position lead to an additional error, for
example a horizontal offset of 19 m over a 0.5 degree slope lead to a vertical offset of ~0.17 m
which is included in all comparisons. The standard deviation between the point-to-point
comparison for Baseline-C with the original (Figure 6a) and the revised star tracker
measurements (Figure 6b) was 4.2 m and 3.8 m respectively, showing that correcting for the
mispointing angle for aberration of light error significantly improves the precision of swath
measurements. While the standard deviation of the point-to-point comparison for Baseline-D





was 3.7 m, showing a slight improvement compared to Baseline-C, which can be attributed to
improved processing of the star tracker measurements documented in Baseline-D.

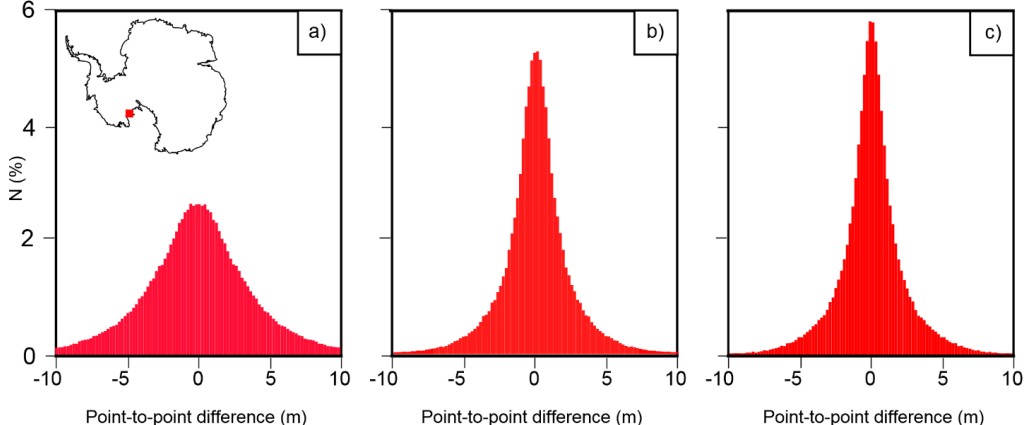


**Figure 6 Point-to-point comparison of swath data over the Siple Dome (red box in map insert) for (a) Baseline-C with**

**original star tracker measurements (b) Baseline-C with revised star tracker measurements and (c) Baseline-D.**


### 3.2.3    SARIn Validation at Austfonna, Svalbard
The Southeastern basin of the Austfonna ice cap, Svalbard, began surging in 2012 (Dunse et
al. 2012; Dunse et al. 2015). The surge resulted in a heavily crevassed surface of the basin,
creating a challenging surface topography for radar altimetry. CryoSat operates in the new and
innovative SARIn mode over the Austfonna ice cap and due to the complex surface, the ice
cap has been chosen as a primary validation site for the CryoSat mission in the ESA CryoSat
Validation Experiment (CryoVEx) and the ESA CryoVal-Land Ice (LI) projects. Based on
recommendations from the ESA project, CryoVal-LI, the 2016 CryoVEx airborne campaign
(Skourup et al. 2018) revised the traditional satellite under-flights to fly parallel lines with
spacing of 1 or 2 km next to the CryoSat nadir-ground tracks. Sandberg Sørensen et al. 2018
used airborne laser scanning (ALS) data collected at Austfonna in 2016 to validate the data
gathered by CryoSat in April 2016, and processed by six dedicated retrackers. We refer the



reader to Sandberg Sørensen et al. 2018 for a detailed description of the applied retrackers and
schematics of the validation procedure. The six retrackers available in the original study were:
(1) ESA Baseline-C L2 retracker (https://earth.esa.int/documents/10174/125272/CryoSat-
Baseline-C-Ocean-Product-Handbook); (2 and 3) The AWI land ice processing, with and
without the use of a digital elevation model (AWI and AWI DEM, (Helm et al. 2014)); (4) The
NASA Jet Propulsion Lab land ice CryoSat processing (JPL, (Nilsson et al. 2016)); (5) The
Technical University of Denmark (DTU) Advanced Retracking System (LARS NPP50,
(Villadsen et al. 2015)); and (6) University of Ottawa (UoO) CryoSat processing  (Gray et al.
2013; Gray et al. 2015; Gray et al. 2017)). All retrackers were applied to the ESA Baseline-C
L1 waveforms.
The geolocation of the SARIn echo is dependent on the phase at the retracking point hence the
geolocated heights, based on different retracker, cannot be directly compared. Sandberg
Sørensen et al., 2018 relied on comparing the precise geolocation of the ALS with the
individual observations from each retracker, and then provided the derived statistics for all
ALS-CS2 crossovers and for the subset of common nadir position for all retrackers. As the
number of common nadir positions will change if new retrackers are added to the study,
Sandberg Sørensen et al. 2018 also provide the validation code as supplementary material to
the publication. Potentially, this code can be used as a benchmark for future retracker
development. Here, we add the April 2016 Baseline-D ice TDS in benchmarking the code to
pinpoint the differences (Figure 7) and highlight improvement in the new Baseline-D. Table 1
provides the updated statistics, (comparable with Table 1 in Sandberg Sørensen et al. 2018).
The addition of the Baseline-D data reduced the number of common nadir positions from 600
to 497. However, when Baseline-C and D solutions are compared, the new baseline improves
the agreement with the ALS observations in Area 2. The results are more mixed in Area 3
where the surface is rougher and heavily crevassed due to the surging behaviour of this area.



However, there is still room for improvement before the dedicated land ice retrackers of AWI,
JPL and UoO are reached.

**Table 1: Updated statistics for Sandberg Sørensen et al. 2018, with the inclusion of the new ESA Baseline-D L2 processing of CryoSat. The improvements of the new processing are especially noticeable in the standard deviation (Std. dev) of observations in Area 2 (see Figure 7).**

| Area | CS2 Data Set | ESA C | ESA D | JPL | AWI (DEM) | AWI | LARS | UoO |
|------|------|------|------|------|------|------|------|------|
| 1 | # of ΔH | 777 (497) | 774 (497) | 725 (497) | 787 (497) | 828 (497) | 768 (497) | 752 (497) |
| | Mean [m] | 2.80 (3.89) | 2.23 (3.83) | 1.14 (-0.06) | 4.65 (3.68) | 4.42 (4.69) | 13.64 (15.45) | 0.93 (0.53) |
| | Median [m] | -1.11 (-1.21) | -1.28 (-1.32) | -0.28 (-0.34) | 2.04 (1.99) | 2.34 (2.28) | 5.53 (5.28) | -0.31 (-0.58) |
| | Std. Dev. [m] | 30.28 (33.60) | 28.58 (34.29) | 11.71 (3.58) | 11.84 (6.59) | 18.45 (18.37) | 43.52 (49.49) | 4.80 (4.53) |
| 2 | # of ΔH | 509 (335) | 507 (335) | 470 (335) | 509 (335) | 512 (335) | 494 (335) | 497 (335) |
| | Mean [m] | -0.76 (-1.40) | -1.54 (-1.69) | -0.48 (-0.49) | 4.31 (1.53) | 2.72 (2.29) | 4.89 (3.84) | -0.56 (-0.76) |
| | Median [m] | -1.04 (-1.07) | -1.24 (-1.26) | -0.34 (-0.52) | 1.63 (1.98) | 2.04 (1.98) | 5.53 (5.01) | -0.97 (-1.10) |
| | Std. Dev. [m] | 14.63 (3.18) | 4.49 (3.34) | 2.93 (1.84) | 12.57 (1.98) | 6.61 (1.98) | 19.19 (21.4) | 1.97 (1.83) |
| 3 | # of ΔH | 268 (149) | 267 (149) | 258 (149) | 278 (149) | 318 (149) | 274 (149) | 256 (149) |
| | Mean [m] | 9.57 (16.23) | 9.39 (16.76) | 4.00 (0.83) | 5.27 (6.20) | 7.15 (6.51) | 29.43 (41.68) | 3.84 (3.39) |
| | Median [m] | -1.43 (-1.90) | -1.80 (-2.01) | -0.01 (-0.23) | 3.78 (3.90) | 3.99 (4.18) | 5.51 (6.46) | 1.54 (1.19) |
| | Std. Dev. [m] | 46.72 (59.37) | 47.45 (60.49) | 18.91 (5.77) | 10.33 (6.22) | 28.35 (6.26) | 65.25 (77.79) | 6.88 (6.92) |




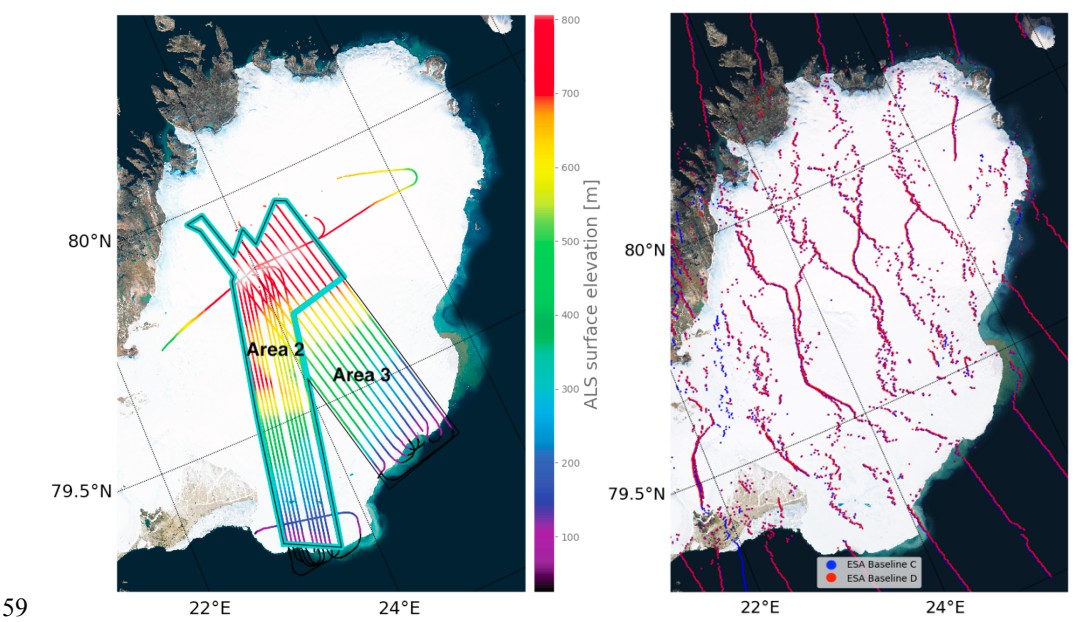

**Figure 7 (Left panel) The surface elevation measured by the CryoVEx airborne laser scanning. The thin black line outlines the entire study area (Area 1); the two subareas are indicated in the figure. Here, Area 3 is covering the complex surface topography of the surging basin of the Austfonna ice sheet. (Right panel) the geolocations of the two ESA L2 Baselines. Map Data ©2019 Google**

### 3.3 Sea Ice

### 3.3.1 Stack Peakiness Implementation

Statistics that describe the power of the stack in CryoSat were already present in the previous Baselines: Stack Kurtosis and Stack Standard Deviation (SSD). While performing an explorative study focused on distinguishing leads from ice surfaces, the adoption of a further parameter was proposed: the Stack Peakiness (SP). This compares the maximum power registered in the Range Integrated Power (RIP) with the power obtained from the other looks. It is also important to notice that this is different from the peakiness of the multi-looked waveform. The latter is influenced by all the looks ("multi-looked"), while the SP compares the influence of the look with the highest power (supposedly at nadir) with the looks taken at



different viewing angles. The advantages in using the SP as a method of discriminating sea ice
floes from leads, instead of (or together with) Stack Kurtosis (SK) and SSD, are described in
Passaro et al., 2018. The evolution of the SP over a sea ice covered area is compared with the
SK and SSD stored in the official product (at the time of Baseline-C). The evolution of SP in
the lead areas are similar: a peak, which corresponds to the strongest return from the zero-look
angle compared to the other looks, is easily identifiable, but the lead returns also influence the
measurements nearby. The lead areas are also characterised by high kurtosis and low SSD, but
these two indices fail to univocally show a local maximum or minimum. The kurtosis presents
multiple peaks, which may be attributed to high power in non-zero look angles due to residual
side-lobe effects; the SSD, being based on a Gaussian fitting, is not able to distinguish subtle
differences in the power distribution of the very peaky RIP waveforms in the lead areas. The
exact formula to compute SP is reported in Passaro et al., 2018. The SP has now been included
in the new Baseline-D.

**3.3.2    CryoSat Baseline-D freeboard assessment**
Previous analyses carried out by the CryoSea-Nice ESA project (https://projects.along-
track.com/csn/) highlighted important over-estimations in the freeboard values of the ESA
CryoSat Baseline-C products relative to in-situ data. Following these conclusions,
modifications have been made to develop the new ESA CryoSat Baseline-D freeboard product.
We present here the first assessments of this updated version.
The freeboard maps in Figure 8 present the evolution between the two Baselines. They
demonstrate that the Baseline-D mean freeboard values have been significantly reduced. Aside
from a mean bias of about 10 cm (see map Figure 8c) the two solutions remain consistent with
each other. The Root Mean Square (RMS) in each 20 x 20 km$^2$ pixel, that can be considered as
an estimation of the noise level distribution, is about 15 cm for the two Baselines.



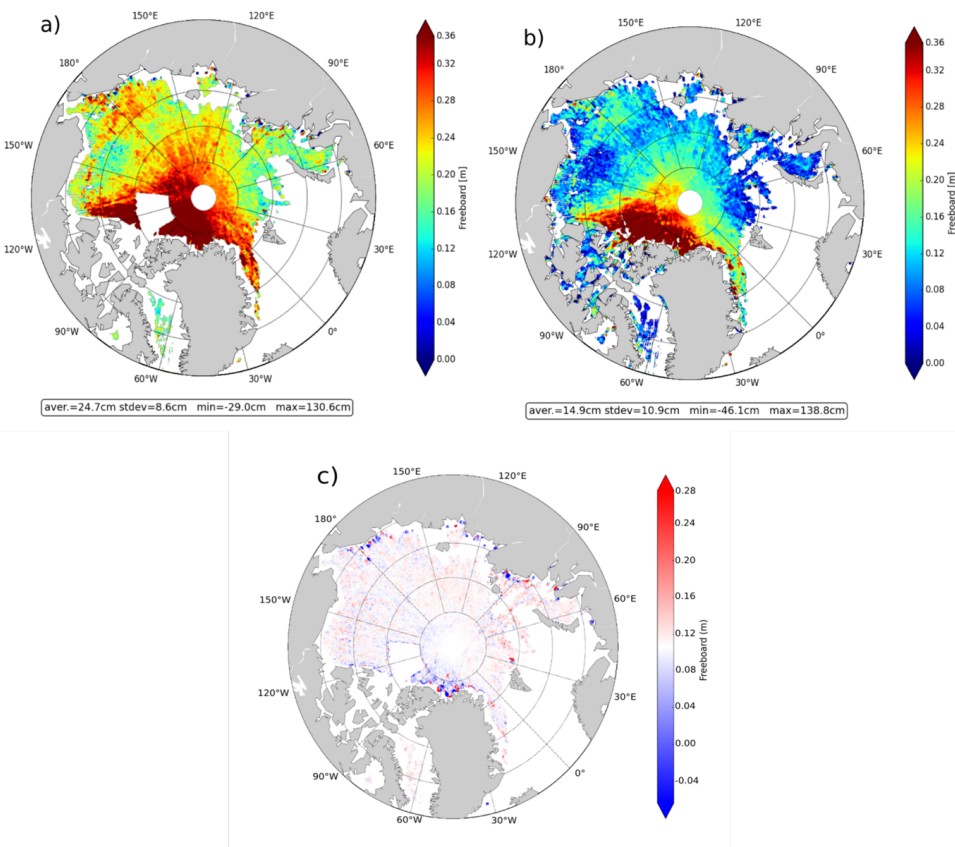


**Figure 8: Monthly freeboard maps from the 10th March 2014 to the 11th April 2014 of the a) Baseline-C and b)**

**Baseline-D versions. The third map c) presents the difference between the 2 previous maps (Baseline-C – Baseline-D).**

**Note that the map c) colour bar is centred on 0,1 m to underline the mean bias deviation between the 2 versions.**


Scatter    comparisons    with    the    Beaufort    Gyre    Exploration    Project    (BGEP,
https://www.whoi.edu/beaufortgyre) and Operation IceBridge (OIB, Kurtz et al., 2013) in-situ
measurements confirm the improvements of the Baseline-D freeboard product quality (see
Figure 9). Relative to OIB, the Baseline-D freeboard mean bias is reduced by about 8 cm,
which roughly corresponds to a 60% decrease. The BGEP data indicate a similar tendency with
a mean draft bias lowered from 0.85 m to -0.14 m (mean draft is ~1 to 1.5 m). For the two in-



situ datasets, the Root Mean Square Deviation (RMSD) is also well reduced from 14 cm to 11
cm for OIB and with a factor 2 for BGEP.

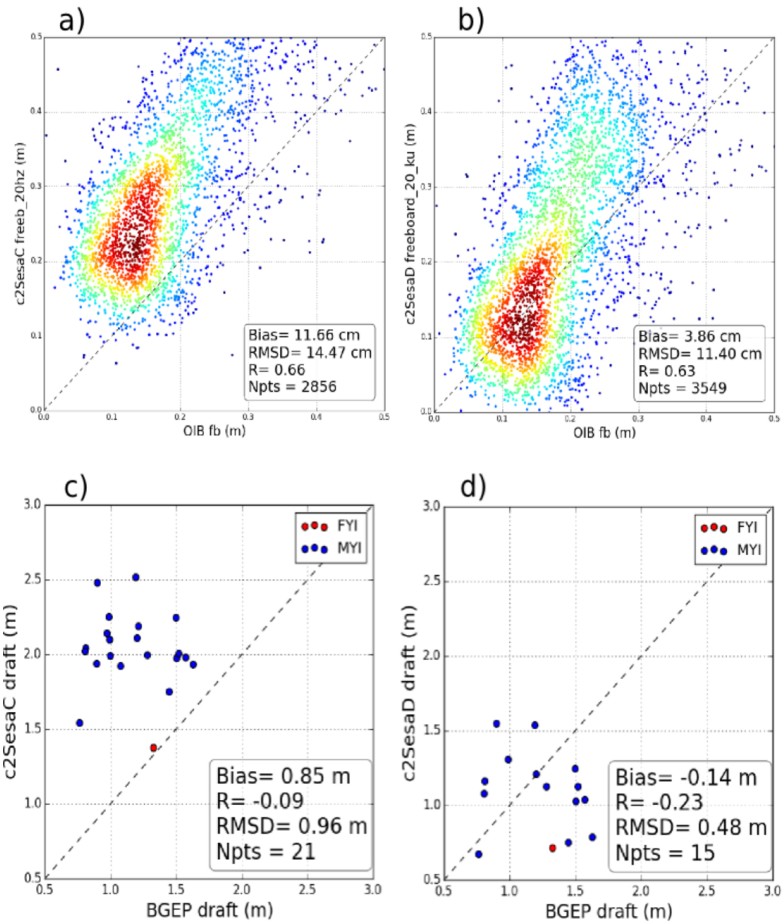


**Figure 9 Illustration of the Baseline-D product improvements by comparison with in-situ measurements. The first**
**two figures compare the 2014 Operation IceBridge (OIB) freeboard measurements with a) the Baseline-C and b) the**
**Baseline-D sea ice freeboard. The two following scatterplots compare the winter 2013/ 2014 Beaufort Gyre**
**Exploration Project (BGEP) sea ice freeboard converted to draft estimations with c) the Baseline-C and d) the**
**Baseline-D sea ice freeboards.**
Some additional comparisons have set the Baseline-D freeboard solution within the range
values of several recent estimations such as Ricker et al, 2014 and Guerreiro et al, 2017. All
together, these results demonstrate the positive improvements of the ESA Baseline-D freeboard



product compared to the previous Baseline-C version. In addition, the improved phase
difference in SARIn mode had positive impacts on sea ice freeboard as presented in the next
section.

### 3.3.3 Impact of SARIn phase difference on freeboard estimation

Satellite altimetry has been used in the last 25 years to estimate sea ice thickness by directly
measuring the sea ice freeboard, i.e., the height of the sea ice above the local sea surface (Laxon
et al., 2003). The different physical characteristics of sea ice and leads, which provide the local
sea surface height, affect the shape and the power of the reflected radar pulses received by the
altimeter, allowing for surface discrimination. Retracking echoes coming from sea ice and
leads enables determination of the height of the sea ice and the sea level, respectively. Finally,
the freeboard height is obtained by subtracting the local sea surface height from the sea ice
elevations.
In sea ice covered regions, the accurate estimation of the sea surface height (SSH) highly
depends on the amount and spatial distribution of leads. A study by Armitage and Davidson,
2014, showed that the CryoSat SARIn acquisition mode can be used to obtain a more precise
SSH, as it enables processing of echoes that are usually discarded because of their ambiguity,
e.g., echoes dominated by the reflection from off-nadir leads. In fact, the phase information
available in the SARIn mode enables the across-track location on ground of the received echoes
to be determined and an off-nadir range correction (ONC) to be geometrically computed,
accounting for the range overestimation to off-nadir leads Laxon et al., 2003. The more precise
SSH obtained from SARIn measurements can reduce the average random uncertainty of
freeboard estimates (Di Bella et al., 2018).
Despite the overall reduction of the random freeboard uncertainty when including the phase
information, pan-Arctic sea ice freeboard estimates from CryoSat Baseline-C SAR/SARIn L1B



products showed large negative freeboard heights at the boundary of the SARIn mode mask
(Figure 10a and Figure 10b). The analysis performed by Di Bella et al., 2019 attributed the
negative freeboard pattern observed in Figure 10a and Figure 10b to large values of ONC,
associated with inaccurate phase differences. The same study determined that the CAL4
correction, responsible to calibrate the phase difference between the signal received by the two
antennas (Fornari et al., 2014), was not applied at the beginning of a SARIn acquisition.
The Baseline-D SAR/SARIn IPF1 applies the CAL4 correction which is closest in time to the
19 bursts of the first SARIn acquisition, improving notably the phase difference and the
coherence at the retracking point. Looking at the Arctic freeboard estimates obtained from
Baseline-D SAR/SARIn L1B products in Figure 10c and Figure 10d, one can notice that the
negative freeboard pattern along the boundaries of the SARIn acquisition mask has
disappeared, highlighting a continuous freeboard spatial distribution throughout the Arctic
Ocean.
The Baseline-D IPF therefore improves the quality of the retrieved heights in areas up to ~12
km inside the SARIn acquisition mask, being beneficial not only for freeboard retrieval, but
for any application that exploits the phase information from SARIn L1B products.




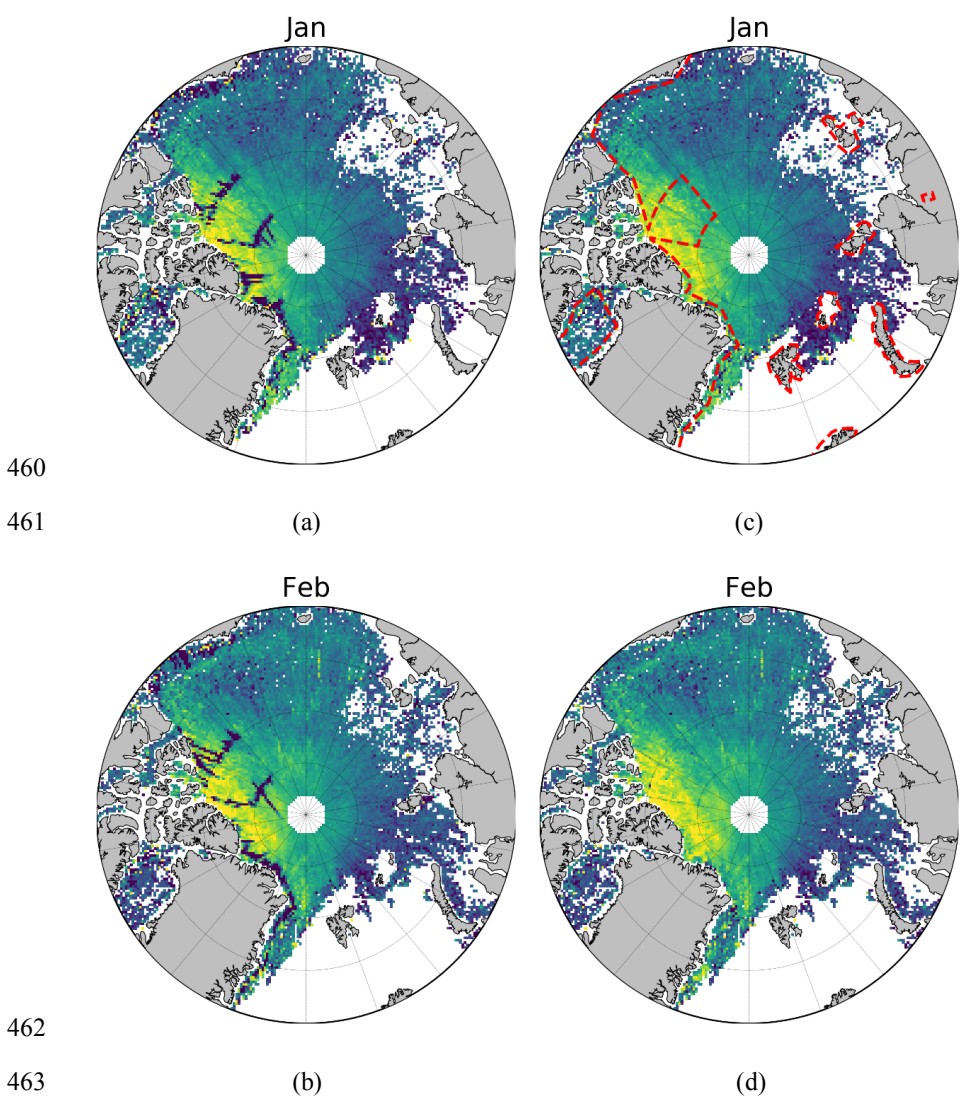

**Figure 10 Gridded monthly freeboard from Baseline-C (a-b) and Baseline-D (c-d) L1b data for the period January/February**

**2014. The dashed red line in (c) represents the boundaries of the SARIn acquisition mask**

### 3.3.4 Impact of algorithm evolution on sea ice thickness consistency

Operational L1B products generated by the CryoSat Baseline-C Ice processor (IPF1C) are a

primary dataset for observing changes sea ice thickness in the northern hemisphere. Examples



for the application of CryoSat L1B products in sea ice climate research are formalised climate
data records such as those of the ESA Climate Change Initiative (CCI) (Paul et al., 2018,
Hendricks et al., 2018b) and the Copernicus Climate Change Services (C3S) (Hendricks et al.,
2018a, Hendricks et al., 2018b). In addition, several agencies and institutes generate sea ice
data records based on the CryoSat L1B Baseline-C products (Tilling et al., 2018), AWI (Ricker
et al., 2014, Kurtz et al. 2014, Kwok et al., 2015, Guerreiro et al., 2017). To estimate the impact
of the algorithm evolution of the CryoSat Ice Processor to Baseline-D (IPF1D) on these sea ice
data records, we compute sea ice thickness ($SIT$) for both IPF1C and IPF1D primary input
datasets with an otherwise identical processing environment. The processing chain for this
experiment has been developed at the Alfred Wegener Institute (AWI) (Ricker et al. 2014) and
we utilise the most recent algorithm version 2.1 (Hendricks et al., 2019). The AWI processor
is implemented in the python sea ice radar altimetry library along with the climate data records
of the ESA CCI and C3S. Processing steps consist of a L2 processor for the estimation of sea
ice freeboard and thickness at full along-track resolution and a L2 processor for mapping data
on a space-time grid for a monthly period with a resolution of 25 km in the northern
hemisphere. For a full description of the algorithm and processing steps we direct the reader to
Hendricks et al., 2019. The CryoSat IPF1D input data is processed with the identical processor
configuration as the current IPF1C based AWI reprocessed product line. The impact analysis
is implemented for 5-month periods of the IPF1D test period (October – November 2013;
February – April 2014) by evaluating pointwise differences (IPF1D – IPF1C) of gridded
thickness from the two CryoSat primary input versions. Monthly statistics of sea ice thickness
differences ($\Delta SIT$) itemised for all grid cells in the northern hemisphere (ALL) as well as for
the SAR and SIN modes of the altimeter is shown in Figure 11 and in Table 2. In addition,
Figure 11 illustrates the regional distribution of $\Delta SIT$ exemplary for the monthly period of
April 2014. The mean monthly thickness difference between IPF1D and IPF1C ($\overline{\Delta SIT}$) varies





between -3 to -15 mm. Its magnitude is increasing over the winter season with highest values
in April, which we attribute to the increase of ice thicknesses over the winter period. However,
the radar mode plays an important role in the $\overline{\Delta SIT}$ result, as thickness measurements from
SAR data are significantly less impacted by the input version than SIN data. Regions with SIN
data therefore drive the magnitude and negative sign for hemispheric $\overline{\Delta SIT}$ (SAR: -5 to 9 mm,
SIN: -17 to -77 mm). On the map in Figure 11 this is particularly visible in the Wingham Box
(WHB), a region where CryoSat has operated in SIN mode from 2010 to 2014 and which has
a higher density of grid cells with negative $\Delta SIT$. The magnitude of $\Delta SIT$ even for SIN is
however small compared to the $SIT$ uncertainty for monthly gridded observations that are
mostly driven by the unknown variability of snow depth, surface roughness and sea ice density.
Average gridded $SIT$ uncertainty in the AWI product for April 2014 is 0.64 m and we therefore
conclude that a maximum $\overline{\Delta SIT}$ of -0.015 m in the period of the TDS is insignificant for the
stability of sea ice data records. This impact analysis however does not provide any insights
into the specific algorithm changes that are causing the observed $\Delta SIT$. We therefore speculate
that the change in power scaling for SIN data between IPF1C and IPF1D is the reason for the
larger impact on SIN data as the AWI surface type classification depends partly on total
waveform backscatter. An update to the surface type classification that includes the additional
stack peakiness information in IPF1D has the potential to further improve surface type
classification and consequently sea ice freeboard and thickness. The AWI processing chain is
based on the python sea ice radar altimetry processing library (pysiral). The source code is
available     under     a     GNU     General     Public     License     v3.0     license
(https://github.com/shendric/pysiral). Reprocessed and operational sea ice thickness with
intermediate parameters for gridded and trajectory products of the AWI processing chain can
be accessed via the following ftp (ftp://ftp.awi.de/sea_ice/product/cryosat2/).


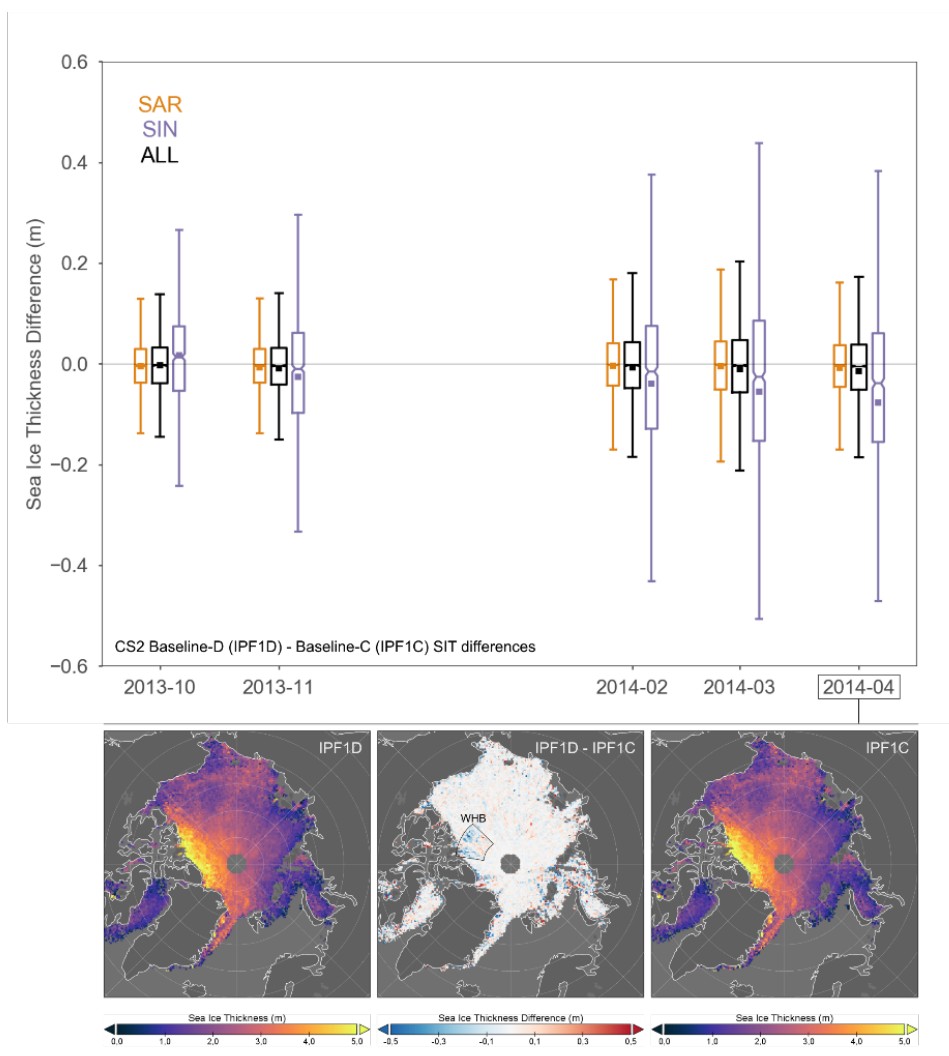


**Figure 11 [Upper panel] Time series of gridded monthly sea-ice thickness difference (ΔSIT) statistics for the AWI sea**
**ice processing chain based on the Baseline-D (IPF1D) test data set and Baseline-C (IPF1C) input. Differences**
**(Baseline-D minus Baseline-C) are colour-coded for all 25 km x 25km grid cells in northern hemisphere (ALL) and**
**separately for SAR and SIN input data. The inner boxed indicates the median difference with the confidence interval;**
**the square marker indicates mean difference ($\overline{\Delta SIT}$) and the vertical line the maximum ΔSIT range. [Lower panel]**
**SIT maps in April 2014 for IPF1D (left), IPF1C (right) and the IPF1D-IPF1C difference (center). The marked region**

**(WHB: Wingham Box) indicates an area where CryoSat operated in SIN mode.**



**Table 2. Mean thickness difference ($\overline{\Delta SIT}$) and standard deviation ($\sigma_{\Delta SIT}$) for all monthly gridded fields during the winter months (October – April) of the Baseline-D TDS. The statistics is broken down into a) all grid cells with data coverage for both baselines b) SAR data and c) SIN data (highest $\Delta SIT$ values).**

|  | SAR+SIN (ALL) | | SAR | | SIN | |
|---|---|---|---|---|---|---|
|  | $\overline{\Delta SIT}$ (m) | $\sigma_{\Delta SIT}$ (m) | $\overline{\Delta SIT}$ (m) | $\sigma_{\Delta SIT}$ (m) | $\overline{\Delta SIT}$ (m) | $\sigma_{\Delta SIT}$ (m) |
| 2013-10 | -0.003 | 0.12 | -0.005 | 0.10 | 0.017 | 0.22 |
| 2013-11 | -0.009 | 0.13 | -0.007 | 0.11 | -0.026 | 0.21 |
| 2014-02 | -0.007 | 0.14 | -0.004 | 0.12 | -0.040 | 0.27 |
| 2014-03 | -0.010 | 0.16 | -0.005 | 0.13 | -0.055 | 0.32 |
| 2014-04 | -0.015 | 0.16 | -0.009 | 0.14 | -0.077 | 0.33 |

### 3.3.5   Lead classification comparison between CryoSat Baseline-C and Baseline-D

Lead classification is essential for retrieving sea ice freeboard and thickness. Previously, the threshold of parameters used for lead classification, such as Stack standard deviation (SSD) and Pulse Peakiness (PP), was re-scaled from Baseline-B to Baseline-C. Lee et al., 2018 proposed a waveform mixture algorithm for lead classification which solely used a normalised waveform. The results of lead classification are the same between Baseline-C and Baseline-D, as illustrated in Figure 12, where the tracks are projected over MODIS imagery. This method stably classifies leads without re-scaling parameters. The lead classification outside of the MODIS image is the same as well. However, as only two example CryoSat products are used for the comparison, this does not guarantee that the results of lead classification are consistent across the entire dataset. The stable lead classification brings a robust retrieval of sea ice freeboard and thickness.



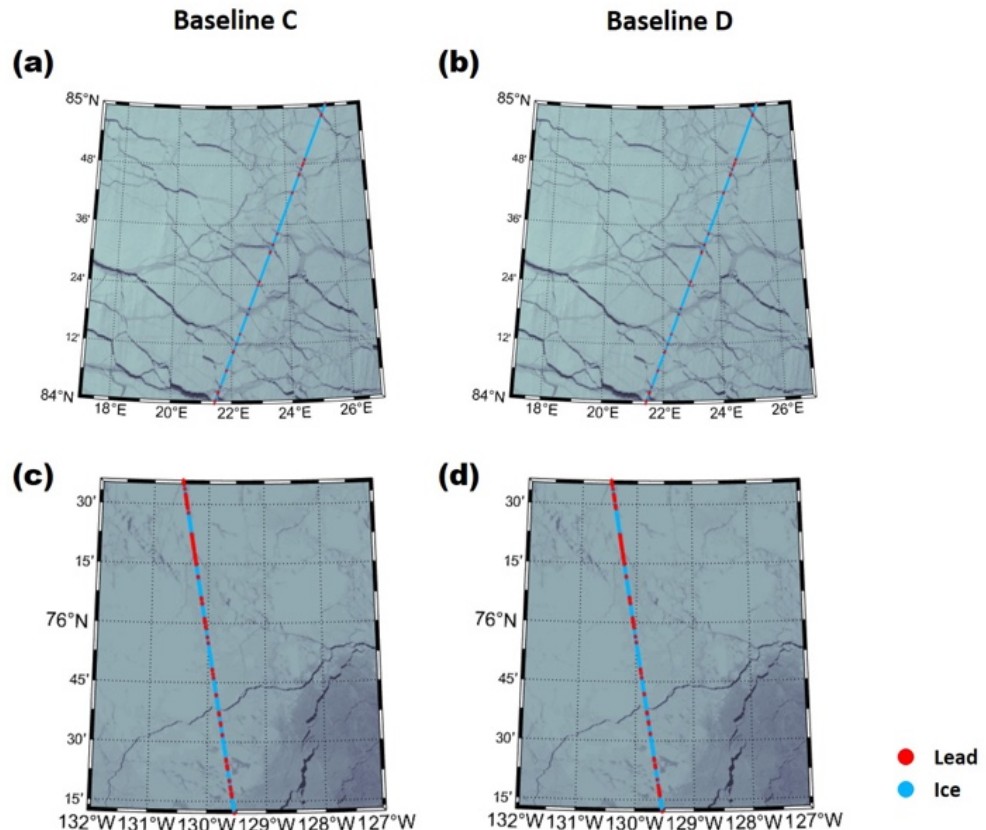


**Figure 12 Red dots represent lead, light blue dots represent ice. (a, b) the MODIS images are from 17 Apr. 2014 22:10**

**(UTC); CryoSat passes over after 21 minutes. (c, d) the MODIS images are from 17 Oct. 2013 22:10 (UTC); CryoSat**

**passes over after 5 minutes.**


### 3.4   Inland Waters

Whilst CryoSat was initially designed to measure the changes in the thickness of polar sea ice,
the elevation of the ice sheets and mountain glaciers, the mission has gone above and beyond
its original objectives. Scientists have discovered that the CryoSat's altimeter has the capability
to map sea level close to the coast and to profile land surfaces and inland water targets such as
small lakes, rivers and their intricate tributaries (Schneider et al., 2017). In this respect, to
evaluate the new CryoSat Baseline-D TDS for lake level estimation two study areas were



selected: Sweden which is covered by SAR mode and the Tibetan Plateau which is covered by
SARIn mode. Both areas have a dense concentration of lakes with a large range of sizes. In
both cases the period September to November 2013 is studied. The evaluated products are the
L2 products (SIR_SAR_L2 and SIR_SIN_L2) for Baseline-C and Baseline-D. The surface
elevations are extracted using a water mask and referenced to the EGM 2008 geoid model. In
the evaluation the standard deviation of the individual water level measurements is estimated
for each track and as a summary measure the median of the distribution of standard deviations
(MSD) is used. The standard deviation is estimated using a robust error distribution consisting
of 70% Gaussian and 30% Cauchy (Nielsen et al, 2015). Furthermore, the percentage of "good
observations" is calculated. Here a good measurement is defined as a measurement within one
meter of the estimated track mean. To get solid statistics only tracks with 15 or more
measurements are used in the analysis. For comparison the analysis was conducted for both
Baseline-C and Baseline-D. For the Swedish area the analysis is based on 26 tracks covering
15 lakes with areas ranging from 29 to 3559 km$^2$. It is found that the MSDs are 7.3 cm and 7.1
cm for Baseline-C and Baseline-D, respectively. With respect to the percentage of "good
observations", a convincing increase is observed for Baseline-D (Figure 13). The larger number
of valid measurements reduces the error of the mean lake level for each track, which is used in
the construction of water level time series. 104 tracks covering 57 lakes with areas between
101 and 2407 km$^2$ are investigated on the Tibetan Plateau. It is found that the MSDs are 19.2
cm and 18.8 cm for Baseline-C and Baseline-D, respectfully. Furthermore, the approximately
60 m offset in the surface elevation that is present in Baseline-C is eliminated in Baseline-D.
For Baseline-D a slight increase in the percentage of "good observations", generally around 5-
10 % for most lakes, is observed.





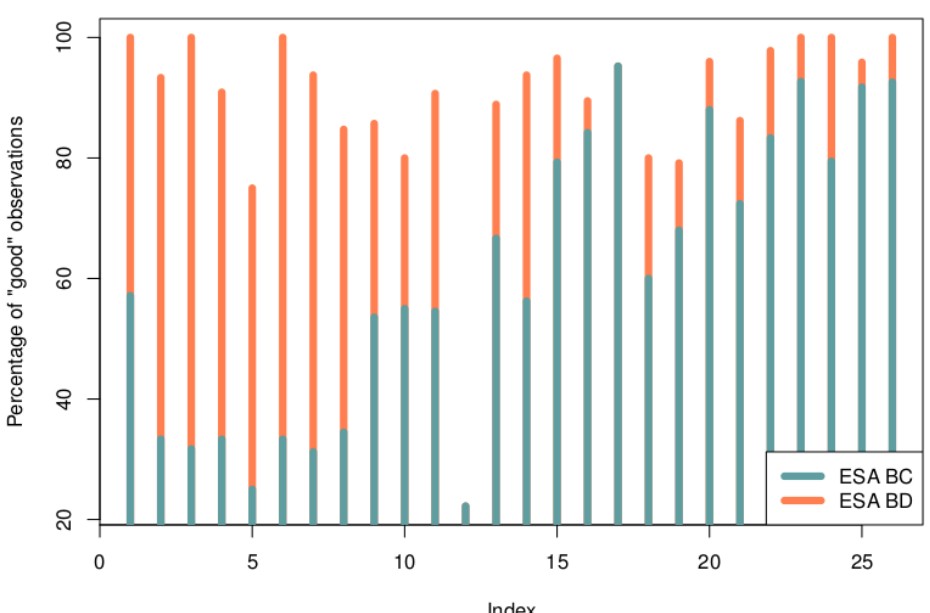


**Figure 13 The percentage of "good measurements" for Baseline-C (blueish) and Baseline-D (coral) based on 26 tracks**
**covering 15 Swedish lakes.**




**4     Conclusions**
In conclusion, validation activities presented in this paper confirm that the new Baseline-D Ice
L1B and L2 data show significant improvements with respect to Baseline-C over all surface
types while the migration to netCDF make these new products more user-friendly than the
previous EEF products. The assessment of a 6-month TDS by multi-thematic CryoSat expert
users was instrumental in confirming data quality and providing an endorsement from the
scientific community before the transfer of the Baseline-D Ice Processors to operational
production on 27th May 2019. The Baseline-D algorithms show significant improvements over
all kinds of surfaces. Most notably, freeboard is less noisy, no longer overestimated and scatter
comparisons with in-situ measurements confirm the improvements of the Baseline-D freeboard
product quality with a reduction of mean bias by about 8 cm, which roughly corresponds to a
60% decrease with respect to Baseline-C. For the two in-situ datasets considered (OIB and
BGEP) the RMSD is also well reduced from 14 cm to 11 cm for OIB and by a factor 2 for
BGEP. In addition, freeboard no longer shows discontinuities at SAR/SARIn interfaces. Over
land ice, the main improvements are due to the increased accuracy in the roll angle. This has
provided better results with respect to previous baseline when comparing the data to a reference
DEM over the Austfonna ice cap region, and improved the ascending and descending crossover
statistics from 1.9 m to 0.1 m. Inland water users also reported significant improvements
including a reduction in previously observed measurement outliers and an increased percentage
of "good observations", generally around 5-10% for most lakes. Overall, this new CryoSat
processing Baseline-D will maximize the uptake and use of CryoSat data by scientific users
since it offers improved capability for monitoring the complex and multi-scale changes in the
thickness of sea ice, the elevation of ice sheets and mountain glaciers and their effect on climate
change.



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
