# Peer review of "CryoSat Ice Baseline-D Validation and Evolutions"

_The Cryosphere, 2019_

## Referee Comment (RC1) · Jack Landy (Referee) · 28 Dec 2019

Review for The Cryosphere Discussions, https://doi.org/10.5194/tc-2019-250

CryoSat Ice Baseline-D Validation 1 and Evolutions Meloni et al., 2019

This study provides a comprehensive evaluation of the CryoSat-2 'Ice' processing chain at Baseline D, including measurements over land ice, sea ice and inland water. Several areas of strong improvement are noted between the Ice processing schemes at Baselines C and D. The conclusions of the study are useful for the large scientific community using Level 1B and 2 observations from the CryoSat-2 Ice processor, particularly those using ESA's official derived land ice and sea ice data products.

The paper is well put together and offers detailed assessment of the validity of Ice measurements at Baseline D. However, I would suggest to the authors to include more

information on the specific changes/evolutions that have been implemented between the Baseline C and D processing chains. I have provided a set of minor comments and recommendations but have no significant concerns with the author's methods or results. My review is focused on the sea ice validation, since that is my area of expertise, although I have made a few minor comments elsewhere.

Please don't hesitate to get in contact if you have questions regarding these comments. Kind regards, Jack Landy

General comments:

1. It is important for tracking the history of each baseline to describe here what issues led to poor quality L2 data in baseline C (e.g. Section 3.3.2) and then what specific modifications were made to the retracking algorithms or processing chains that have led to vast improvements at baseline d.

Minor comments/edits: Line 40-41. Reword to explain why the 12 km is relevant. L48-49. Are the exact same set of auxiliary measurements used for this ice draft analysis at baselines C and D? Fig 1. Please include product acronyms in the captions. Section 1. It would be useful here to include some introduction to the observations produced in the L2 data product. What specific measurements are provided by the ice processor at L2 for land ice, sea ice and lakes? L 140-141. How can the SARIn mode be used to reduce uncertainty? L 143. Need to explain what is meant by 'bad phase difference calibration'. L150-153. What are these parameters for and how Cn they be used by the community? L159. OK to refer to another study, but you need to at least include a definition here of this correction. L170. What is specific about the SARIn mode retracking? Specific in comparison to SAR mode? L172-173. Define retracking before this discussion. You also need to include details of this retracker and how it is implemented. L176. 'Records' is quite ambiguous. Returns? L214-215. This was an issue with baseline c data, or just an issue with the selected TDS for baseline d? L238-239. Clarify whether the angular correction is implemented by the data provider

for baseline d L1B products? Can you explain in a little detail here the source of the angular error and its spatiotemporal dependence? L247. What are these retrackers? What are their differences? It would be extremely useful generally for the altimetry ice community if the authors could provide a table here with details of all the retrackers implemented for each surface type and sensing mode. L249. Citations? L250. Updated surface mask derived from what? By whom? Fig 3. Include an inset map of the location. L327-328. Explain why. L349-350. Add explanation on the latest ESA baseline d retracking algorithm and processing chain. Does it follow one of the other group's processing chains? Are the retracking solutions from other group's algorithms available in the baseline d L2 ice processor data product? L375-376. Clarify. L385.The hyperlink doesn't seem to work. L394. Is this correct? I expect this rms measure is a convolution of the noise with valid signal at the sub grid-cell level. A better estimate for the noise distribution would be obtained from along-track rms of height observations over smooth level ice. Fig 8. Very difficult to see the difference map. Can you enlarge the points and ensure the color scale is cantered so that white = zero. Almost impossible to visualize the positive anomalies here. L404. You need to explain in detail the processing changes that have led to such extreme improvements here. Fig 9a and b. Please include the best-fit line so the reader can see the deviation from 1:1. How were the OIB freeboard observations processed? Are they an official NSIDC product? How are the CS2 observations converted to draft from freeboard? Most importantly what assumptions were made about the snow load? L422-42. This paragraph seems more appropriate for the introduction. L438-439. By what degree can it be reduced? I would also expect it to reduce systematic uncertainty associated with biases in the SSH retrieval. L471-472. Most of these citations do not correspond with the AWI data product. L479-480. So is the ESA retracking algorithm used to derive freeboard, then the remaining processing uses the AWI chain? Or is the full processing from AWI but using different data baselines? L489. Is the phase used to produce an ONC in this processing chain? L504-508. This passage requires explaining in more detail. L536. Identical? Fig 12. It is very difficult to observe any differences between these classifications if indeed there are any. If there are, can you use extra panels to highlight the differences? L562-563. Explain. L564-565. Why is one meter considered to be good? Do you mean the lake mean height from a single track? L575. Why was such a large offset present at baseline c?! Fig 13. Is this a stacked bar chart? If not, move the BD bars next to each BC bar. L590. The lower noise level is not really confirmed here, as I explained in the comment above this would require a different approach to ascertain. L599. Which statistic? Mean bias, rmse..?
* * *

---

## Referee Comment (RC2) · Eero Rinne (Referee) · 8 Jan 2020

Major comments:

The manuscript provides a good overview of the improvements in Cryosat-2 Baseline D product over the past Baseline C. This is an important paper for anyone using the product and deserves to be published. Overall quality of the paper is good. However, in my opinion, the authors should include more of the details of the processing in the manuscript and not only link to ESA technical notes online. Furthermore, there are missing details on some of the comparisons – for example the CS2/BGEP comparison is lacking information on averaging altogether. Also, one of the subsections (3.3.5, lead detection) needs considerable work.

The Section 3.3.5. is the weakest of the manuscript and must be considerably improved before publication. I would expect to see the justification of the inclusion of stack peakiness and the new classification scheme would be better than the old one. However, what is now shown is that the surface type classification of Baseline C and Baseline D are identical and for that only two tracks over two MODIS images are used. What should be (at least) included here is an overall statistical analysis of surface classification results over the whole Arctic (in the style of Figure 11) to see if the two are really same. If that is the case, the authors should further discuss why the SP was used in the classification. If the two classification results are different, the authors should make a solid point why the one in Baseline D is better than the Baseline C version.

Minor comments:

Section 2 - Please include a short list (or a table) of the main variables in L1b and L2 products and what are their expected uses. Yes, they are in the product handbook, but they deserve to be mentioned in this paper as well.

L122-123 Perhaps this (reasonably short) table could be included as an annex to this paper as well?

Section 2.2. Here I would love to see a statement if there were anomalies or problems with Baseline-C that are still not fixed in Baseline-D. Maybe everything is fixed, but I'd love to know if there are pending improvements left.

L172-175 The new retracker should be described in detail. The authors should also present the rationale of choosing the retracker.

L182 "Some tuning of the thresholds for the other metrics" - please tell us what kind of tuning and on which metrics!

L189 Why not just tell us what the surface type mask model now is?

L241 typo – were compute

L250 – Where does this mask come from? Which mask is it?

L253-L259 Also the retracker has been changed, has it not? How can we distinguish

the effect of new retracker and new slope correction?

L381 I would love to see the formula here as well. As well as a detailed description how it is used in the surface classification process. Even the Design Summary document does not include the thresholds used – and they might be beneficial for anyone trying to improve the surface classification in the future.

Section 3.3.4 – In addition to WHB, there are also significant differences in the Kara and Barents seas. Would be good to mention and discuss there in the text. I would reckon this has something to do with relatively thin ice and lot of specular echoes in the area. Maybe include a zoomed version of Figure 11 difference map for these areas as well?

L400 – 413 – Which ice type (density) and snow estimates are used in Baseline D? Are they same as in Baseline C? How is data averaged (both spatially and temporally)? Are all BGEP moorings used? Averaged together? What about OIB -which OIB freeboard is used here: radar or laser? How close to each other CS2 and OIB points need to be in place and time to form a pair? Averaging?

L409-413 – The caption is confusing. What it should say is C and D are BGEP drafts compared to drafts calculated from CS-2 freeboards.

L559 – which water mask?

L564 – Why one meter? Where does this definition stem from? How would results change of more strict requirement (say 50 cm) would be used?

L575 – where did this offset originate from and which correction fixed it?

L590 - "All kinds" however limited here to land ice, sea ice and (marginally) inland water. Rephrase.

---

## Referee Comment (RC3) · Anonymous Referee #3 · 13 Jan 2020

Review of 'CryoSat Ice Baseline-D Validation and Evolutions' in The Cryosphere Discussions
10th January 2020

This article is concerned with the most recent CryoSat-2 processing and dataset version, Baseline-D, which has been operational since May 2019. The paper provides an overview of the main updates and improvements since the previous Baseline-C version, both at Level-1B and Level-2 stages. The discussed improvements at L1B stage are: Transition to the more ergonomic NetCDF file format from EEF; the eradication of anomalously negative radar freeboards at the SAR/SARIn mode boundary; the inclusion of two additional stack parameters; and inclusion of the USO correction to the window delay parameter. Improvements at the Level-2 stage are; transition to NetCDF format; the inclusion of sea ice freeboard data for SARIn mode; a new retracker for diffuse waveforms; improved surface type discrimination with the implementation of the Stack Peakiness parameter; implementation of new slope models for land ice elevation correction; and the inclusion of some additional parameters in the L2 files. The second half of the manuscript offers a series of land ice and sea ice comparisons/validations, either comparing the data with itself (e.g. during ascending and descending passes), with previous Baseline-C data, or with independent observations.

Given that the move to Baseline-D has already happened, it is important that the community understand the main changes since the previous baseline and are convinced that the new data is at least consistent if not improved. This paper offers some important findings to this end but the structure and writing need improving. I therefore recommend the paper's publication subject to the following revisions.

Main points:

1. The paper suffers some continuity issues, where sections can feel a bit disjointed and the use of terminology is not consistent throughout. In particular, the following points should be addressed:
   - I find the subsections of the land ice section (3.2) quite confusing. Section 3.2.1 'Impact of algorithm evolution on land ice products' includes different case examples over East Antarctica and Austfonna. The following sections (3.2.2 and 3.2.3) are then concerned with swath data over Antarctica and SARIn data over Austfonna. Why do these two sections not also fall under 'Impact of algorithm evolution on land ice products'? Perhaps Section 3.2.1 could be broken into a number of subsections, each with a different case example, including sections 3.2.2 and 3.2.3? Also it may flow better if the cases over Austfonna followed each other.
   - Move the first paragraph of section 3.3.3 to the beginning of section 3.3.2. Should section 3.3.3 go inside section 3.3.2 since it falls under Baseline-D freeboard assessment?
   - In the abstract and elsewhere, the processors are referred to as 'Ice Baseline-C' and 'Ice Baseline-D'. However in section 3.3.4, 'IPF1C' and 'IPF1D' appear for the first time and are used throughout this section. Please choose a name/acronym for the processors, define them in the introduction, and ensure their use is consistent throughout the manuscript.
   - As it stands, the relevance of section 3.3.5 is hard to appreciate. The section refers to re-scaling of parameters between baselines B and C, but makes no mention to the re-scaling of parameters from Baseline-C to Baseline-D, which was presumably necessary and would be of interest to the reader. Since the L2 Baseline-D processor does not use the Lee 2018 method for lead identification, more explanation is needed to tie this section in and relate it to the previous content.

2. The article heavily cites documents (mainly ESA documents) via URLs in the text, some of which feel like necessary supplementary material to the main text (e.g. the 'CryoSat-Baseline-D-evolutions' document). Is the permanence of these URLs certain? If not - can they be put on a DOI or provided in a supplementary?

3. Page 13, line 245 - You say that "Most of the parameters were found to show a close agreement.." I find this quite vague and expect users would want to know more on how parameters compare between each baseline. Could you include more details or a table?

4. The validation of LRM data over land ice depends on a comparison to REMA. Please could you provide some details about how REMA is built, e.g. what data is used in its construction, and a justification for why you chose to validate with REMA? Is there any particular reason that this area of East Antarctica was chosen?

5. Why did you not validate/compare Baseline-D over land ice with an independent observation dataset like IceBridge?

Minor points

**Section 1:**
- Page 2, line 27 - "on 8 April 2010" -> "on the 8th April 2010"
- Page 2, line 30 - Is CS2's repeat cycle not exactly 369 days with a 30-day sub cycle? Not sure what 'quasi' is referring to here.
- Page 2, line 50 - "with a factor 2" -> "by a factor 2"
- Page 3, line 53 - "CryoSat ice Baseline-D" -> "CryoSat Ice Baseline-D"
- Page 4, line 71 - "affecting the Polar Regions" -> "affecting Earth's polar regions"
- Page 4, line 78 - "which are contributing to global sea level rise." -> "which influence global sea level." (variations in thickness can mean thickening which does not cause sea level rise)
- Page 4, line 83 - "interferometric" -> "Interferometric"
- Page 4, line 88 - "The ice products are currently generated with the Ice Baseline-D processors". Please be more specific here. Since when are they generated with Baseline-D processors? covering which operational period?
- Page 5, line 106 - "sea ice and inland waters domains." -> "sea ice and inland waters."

**Section 2**
- Page 7, line 139-141. Here you mention the findings of Armitage et al. 2014 but they do not relate directly to the next sentence. This sentence could be removed since you discuss the relevance of the Armitage and Davidson study to the anomalously negative freeboards in section 3.3.3
- Page 8, line 147. Please explain what a Doppler cell is and reference a paper e.g. Raney 1998, rather than an impermanent URL.
- Page 8, line 148-153. Consider merging these two sentences: "At Baseline-D, two additional stack characterisation parameters (also known as Beam Behaviour Parameters) have been added to the SAR/SARIn L1B products: i) the stack peakiness (Passaro2018), which can be useful in improving sea ice discrimination, and ii) the position of the centre of the Gaussian fit......."
- Page 8, line 167 - "the freeboard sea ice processing" -> "the sea ice freeboard processing"
- Page 9, line 169 - "The height value…" - I don't know what height value you are referring to, could you be more specific?
- Page 9, line 172 - Could you detail briefly this new retracker? Is it physical / threshold etc.
- Page 9, line 178 - "based on the peakiness of SAR waveforms" - add "(see section 3.3.1)"
- Page 9 , line 187 - "..to make the correction more responsive.." - What correction? Please be more specific.
- Page 10, line 194 - "in addition to the height". What height?

**Section 3**
- Page 11, line 210 - "…data files was" -> "…data files were"
- Page 11, line 212-213 - "…geophysical corrections were checked to ensure that they were computed correctly". This is a little vague, can you say something more concrete about how they were checked?
- Page 12, line 222 - "…generated using the Baseline-C are…" -> "…generated using the Baseline-C processors are…"
- Page 12, lines 225-229 - Consider moving the sentence starting "To derive mass balance.." to after the sentence ending "..should help to reduce this uncertainty." on line 239 to aid the flow of this section.
- Page 12, line 231 - "respectively mass change" - I don't know what 'respectively' means here.
- Page 12, line 234-236. Please provide a reference for this statement.

- Page 12, line 236-237 - "However, a small attitude angle error interpreted as a mispointing error has been observed.." Observed by who? Please provide a reference.
- Page 12, line 241 - "were compute" -> "were computed"
- Page 12, line 242 - "Level 2 "in depth" (L2I) product retracker" - what is this? Is there a technical note or article you could reference?
- Page 13, line 247 - are the constant offsets on Sigma0 you list for Baseline-D minus Baseline-C or vice versa?
- Page 13, line 247-249. The sentence starting "This needs to be considered…" does not make sense, please re-phrase.
- Page 13, line 249-250 - "Furthermore, Baseline-D uses an updated surface type mask. This…" -> "A new surface type mask has been implemented in Baseline-D, significantly improving resolution in the ice shelf area…."
- Page 13, Figure 2. Please include a scale e.g. "Orange=Ice shelf, Blue=Ice sheet".
- Page 13, Figure 2 caption: "Ronne ice shelve" -> "Ronne ice shelf"
- Page 14, line 255 - "This slightly changes the LRM slope corrected elevation". What does slightly mean? 1% ? 10% ? Please quantify the change.
- Page 14, line 256 - "… for a large area in East Antarctica…". Can you explain why you chose this area? Please also include a map of Antarctica or East Antarctica to show where this region is.
- Page 14, Figure 3. Please say in the caption what the numbers are, i.e. "Mean REMA-CS2 difference= +0.13 ± 1.2 m" etc
- Page 14, line 258 - "Differences to an independent Antarctic elevation model…" -> "Differences between slope corrected elevation and an independent Antarctic elevation model…"
- Page 14, line 259 - "The differences vary spatially and the overall mean…" -> The differences vary spatially and the overall mean difference (REMA minus CS2)…"
- Page 14, line 265 - "…however major improvements" -> "…however offers/implies major improvements"
- Page 14, line 266 - "…swath data processed for ascending and descending tracks" - For what period?
- Page 15, line 268 - "The large positive anomaly is a known.." -> "The large positive anomaly (blue area in Fig. 4) is a known.."
- Page 15, line 271 - "(subpanel B) could be reduced". 1) No subpanels are labelled in the figure. 2) I don't know what you mean by "could be reduced"
- Page 15, Figure 4:
    1. Please add labels "Baseline-C" and "Baseline-D" on the right-hand side and "Ascending" and "Descending" above the sub-panels.
    2. Please make the labels of the colour bar larger.
    3. In the caption please change "Differences to relative elevation model.." to "Differences between CryoSat elevation and reference elevation model…" or "Deviation of CryoSat elevations from reference elevation model…"
- Page 16, Figure 5. "Crossovers between ascending descending.." -> "Difference in elevation between ascending and descending crossovers…"
- Page 17, line 291 - "a point-to-point comparison was performed". Please make clear in the text that you are not comparing Baseline-C points with Baseline-D points as this is how this reads.
- Page 17, line 310 - "…which is included in all comparisons." Do you mean that it is accounted for in the comparison, i.e. subtracted?
- Page 18, line 322-323 - "CryoSat operates in the new and innovative SARIn mode…" -> "CryoSat operates in SARIn mode…"
- Page 18, line 326 - "recommendations from the ESA project, CryoVal-LI, the 2016…" -> "recommendations from CryoVal-LI, the 2016…"
- Page 19, lines 333-339. "The AWI land ice processing", "NASA JPL land ice processing" and "University of Ottawa CryoSat processing", are these retrackers? or are they processors? Please tidy these distinctions up in the text.
- Page 20, "…before the dedicated land ice retrackers of AWI, JPL and UoO are reached." I don't know what a retracker being reached means.
- Page 20, Table 1. Please change "Mean [m]", "Median [m]" and "Std. Dev. [m]" to "Mean ALS-CS2 difference [m]", "Median ALS-CS2 difference [m]", "Std. Dev. on ALS-CS2 difference [m]". Also state in the caption what the numbers in the brackets represent.
- Page 21, Figure 7. In caption: "CryoVex airborne laser scanning." -> "CryoVex Airborne Laser Scanner (ALS)."

- Page 21, line 362 - "Statistics that describe the power of the stack in CryoSat were…." -> "Statistics that describe the power of the CS2 waveform stack were…."
- Page 21, line 365-366 - "This compares the maximum power registered in the Range Integrated Power (RIP) with the power obtained from the other looks". The RIP of which look? From Passaro, I understand that the Stack Peakiness "compares the power at the zero look angle with the backscatter registered in the other looks", please check your definition.
- Page 21, line 369 - "..with the highest power (supposedly at nadir) with the looks". Again, it is the power in the nadir beam that is compared with the off-nadir looks. I understand that the RIP waveform is first normalised by its peak value- which may not be at nadir- but this sentence confuses the two steps.
- Page 22, line 372 - "The evolution of the SP over a sea-ice covered area" - I don't know what you mean by 'evolution' here. Evolution in time?
- Page 22, line 382. Could you re-iterate at the end of this section that the SP parameter is implemented in lead discrimination for L2 sea ice products (as discussed in section 2.2) and mention the thresholds that are used or direct reader to where they can find the thresholds.
- Page 22, line 386 - "…highlighted important over-estimations in the freeboard values of the ESA CryoSat Baseline-C products relative to in-situ". Is there a reference to support this claim? The URL is just a link to the site for CSN.
- Page 22, line 390 - "…Figure 8 present the evolution between two Baselines." I think this figure is simply showing the difference rather than any evolution.
- Page 22, line 392-393 - "…the two solutions remain consistent with each other". Could you add a comment on the larger differences in the MYI region north of Greenland?
- Page 22, line 393 - "The Root Mean Square (RMS) in each…" Do you mean the Root Mean Square deviation from the average value in each pixel? i.e. the standard deviation?
- Page 23, line 401. Please state which OIB dataset was used - Quicklook / L2 / L4?
- Page 24, line 407 - "…with a factor…" -> "…by a factor…"
- Page 24 line 414-415. I don't know what this sentence means.
- Page 25, line 418 - "…SARIn mode had positive impacts on sea ice freeboard". The word 'positive' here is ambiguous - do you mean positive as in good? or positive as in greater than zero? Please clarify in the text.
- Page 25, line 437. Why is Laxon 2003 referenced here? Laxon does not mention off-nadir leads.
- Page 26, line 446 - "..responsible to calibrate.." -> "..responsible for calibrating…"
- Page 28, line 471 - Why is AWI listed here but none of the other groups?
- Page 28, line 489 - "..SAR and SIN modes of the altimeter is shown…" -> "..SAR and SIN modes of the altimeter are shown…"
- Page 29, line 492 - "Its magnitude is increasing.." -> "Its magnitude increases.."
- Page 29, line 505 - "We therefore speculate that the change in power scaling for SIN […] is the reason…". Please provide further details about this change in power scaling as it's unclear what you are referring to here.
- Page 30, Figure 11 caption: "The inner boxed indicates.." -> "The inner box indicates.."
- Page 32, line 551 - "…discovered that the CryoSat's altimeter.." -> "…discovered that CryoSat's altimeter.."
- Page 32, line 574 - "respectfully" -> "respectively"
- Page 35, line 597 - "with respect to previous baseline" -> "with respect to the previous baseline"

---

## Author Comment (AC2) · 18 Feb 2020

**Answers to Referee 2**

Major comments:

The manuscript provides a good overview of the improvements in Cryosat-2 Baseline D product over the past Baseline C. This is an important paper for anyone using the product and deserves to be published. Overall quality of the paper is good. However, in my opinion, the authors should include more of the details of the processing in the manuscript and not only link to ESA technical notes online. Furthermore, there are missing details on some of the comparisons – for example the CS2/BGEP comparison is lacking information on averaging altogether. Also, one of the subsections (3.3.5, lead detection) needs considerable work.

The Section 3.3.5. is the weakest of the manuscript and must be considerably improved before publication. I would expect to see the justification of the inclusion of stack peakiness and the new classification scheme would be better than the old one. However, what is now shown is that the surface type classification of Baseline C and Baseline D are identical and for that only two tracks over two MODIS images are used. What should be (at least) included here is an overall statistical analysis of surface classification results over the whole Arctic (in the style of Figure 11) to see if the two are really same. If that is the case, the authors should further discuss why the SP was used in the classification. If the two classification results are different, the authors should make a solid point why the one in Baseline D is better than the Baseline C version.

**Reply:** We have entirely re-written section 3.3.5 adopting stack peakiness in the lead classification. This lead classification using SP conservatively returns fewer leads than previous lead classification, including SSD and PP (Tilling et al. 2018). We added a comparison in the monthly lead fraction map in April 2011. While overall spatial patterns are similar, the mean lead fraction in the whole Arctic is different. The lead classification using SP identifies somewhat big and wide leads with over SP 15 (Fig. 1). The threshold of SP should be optimised by evaluating the accuracy of ice freeboard and thickness. Adopting SP might consequently improve ice freeboard and thickness estimation by isolating nadir returns. Although it is hard to draw firm conclusions from this comparison, it is expected that adopting stack peakiness might help isolate nadir returns.

Minor comments:

Section 2 - Please include a short list (or a table) of the main variables in L1b and L2 products and what are their expected uses. Yes, they are in the product handbook, but they deserve to be mentioned in this paper as well.

**Reply:** Thanks to the referee for the suggestion. The following paragraphs will be added in the revised version of the manuscript in section 1 in correspondence with figure 1 which explains the actual implemented processing steps.

"The CryoSat Ice Processor generates Level 1B and Level 2 Ice products from L0 LRM, SAR and SARIn products. These products are primarily designed for the study of land ice and sea ice, although they are also relevant and useful to a wide range of additional applications.

Level 1B data consist, essentially, of an echo for each point along the ground track of the satellite. In all three modes, the data consists of multi-looked echoes at a rate of approximately 20 Hz.
Level 2 products instead are considered to be most suitable for users, as they contain surface height measurements fully corrected for instrumental effects, propagation delays, measurement geometry and additional geophysical effects such as atmospheric and tidal effects. In the L 2 products, the value of each geophysical correction provided is the value applied to the corrected Surface Height. Sea level anomalies and radar freeboard data are also included in the CryoSat Level 2 data products"

L122-123 Perhaps this (reasonably short) table could be included as an annex to this paper as well?

**Reply:** the link reported is related to an official ESA document and according to the authors, it is more appropriate to refer to the official document instead to copying the table in the actual manuscript.

Section 2.2. Here I would love to see a statement if there were anomalies or problems with Baseline-C that are still not fixed in Baseline-D. Maybe everything is fixed, but I'd love to know if there are pending improvements left.

**Reply:** All the foreseen evolutions and fixes of Baseline-C, have been implemented in the current Baseline-D processing chains (L1B + L2). Obviously, there is always room for improvement in operational products such as the CryoSat ones. Any other improvements or evolutions suggested by the scientific community will be analysed and considered by ESA to be potentially implemented in a future version of the ice processing chain.

L172-175 The new retracker should be described in detail. The authors should also present the rationale of choosing the retracker.

**Reply:** the sentence will be changed to:

"In addition, a new threshold-of-first-maximum retracker is used…"

And after that sentence, the following text will be added:

"Retracking is the process whereby the initial range estimate in the L1B data is corrected for the deviation in the first echo return within the waveform from the reference position."

L182 "Some tuning of the thresholds for the other metrics" - please tell us what kind of tuning and on which metrics!

**Reply:** Some tuning of the thresholds for the other metrics has also been performed, based on analysis of the test datasets

In addition, the following text will be added:

The discrimination algorithm currently uses sea ice concentration, waveform peakiness, and standard deviation of the stack of waveforms as metrics, in addition to peakiness of the stack. The discrimination thresholds are checked and adjusted whenever the L1 processing is modified to maintain the discrimination results.

L189 Why not just tell us what the surface type mask model now is?

**Reply:** The Level 2 products contain a flag word, provided at 1 Hz resolution, to classify the surface type at nadir. This classification is derived using a four-state surface identification grid, computed from a static Digital Terrain Model 2000 (DTM2000) file provided by an auxiliary file to the processing chain.

L241 typo – were compute

**Reply:** the word "were" is related to "L2 type products".

L250 – Where does this mask come from? Which mask is it?

**Reply:** The Level 2 products contain a flag word, provided at 1 Hz resolution, to classify the surface type at nadir. This classification is derived using a four-state surface identification grid, computed from a static Digital Terrain Model 2000 (DTM2000) file provided by an auxiliary file to the processing chain.

L253-L259 Also the retracker has been changed, has it not? How can we distinguish the effect of new retracker and new slope correction?

**Reply:** We didn't find any differences in the retracked range for LRM retrackers. Therefore, the effect of slope correction is independent of retracked ranges and can be distinguished.

L381 I would love to see the formula here as well. As well as a detailed description how it is used in the surface classification process. Even the Design Summary document does not include the thresholds used – and they might be beneficial for anyone trying to improve the surface classification in the future.

**Reply:** Empirical thresholds are found in Passaro et al. 2018. Given the complexity of the analysis and the length of its description, we do not find the scientific value of listing here the procedure which is already described in another peer-reviewed scientific paper.

Section 3.3.4 – In addition to WHB, there are also significant differences in the Kara and Barents seas. Would be good to mention and discuss there in the text. I would reckon this has something to do with relatively thin ice and lot of specular echoes in the area. Maybe include a zoomed version of Figure 11 difference map for these areas as well?

**Reply:** We agree that there are differences in Kara and Barents seas. We however did not highlight this region, as the observed difference is not related to a change in the IPF1C and IPF1D algorithms but rather to a lower number of orbit data sets in the IPF1D test data set. We have clarified this now in the text:

*Average gridded SIT uncertainty in the AWI product for April 2014 is 0.64 m and we therefore conclude that a maximum ΔSIT of -0.015 m in the period of the TDS is insignificant for the stability of sea ice data records. This bias also includes an issue in the Barents and Kara Seas, where the number of orbits in the IPF1D test data set was less than in the IPF1C data and minor thickness differences can be observed in Figure 11 due to this selection bias.*

L400 – 413 – Which ice type (density) and snow estimates are used in Baseline D? Are they same as in Baseline C? How is data averaged (both spatially and temporally)? Are all BGEP moorings used? Averaged together? What about OIB -which OIB freeboard is used here: radar or laser? How close to each other CS2 and OIB points need to be in place and time to form a pair? Averaging?

**Reply:** These are indeed very good questions. Within the CryoSat Baseline D products the freeboard is a radar freeboard. Therefore, it does not require neither ice and snow densities nor snow depth.

For our validations we need these information to convert the radar freeboard into sea ice thickness or draft. For that purpose, we use the density provided by Warren99 with the official OSI SAF ice type product available on the NSIDC to separate the FYI and the MYI. The snow depth used to take into account the decrease in radar velocity in the snow pack is the same Warren99 modified climatology for the 2 baselines. All the BGEP mooring measurements of the 2013-2014 winter are used to perform the comparison (specified in the figure label). The OIB dataset used is the NSIDC Quicklook version available at https://daacdata.apps.nsidc.org/pub/DATASETS/ICEBRIDGE/Evaluation_Products/IceBridge_Sea_Ice_Freeboard_SnowDepth_and_Thickness_QuickLook (This point is added L401)

To process the OIB freeboard, we use the difference between the ATM laser total freeboard and the snow depth of the snow radar. The exact methodology will be added into the article (L400).

The same methodology is used for OIB and BGEP. These in situ data are gridded into monthly EASE2 500*500 grids (the same grid as for the altimetric freeboard product). Each in situ 'measurement' shown in figure 9 is the average of all data in a 12.5 km x 12.5 km pixel size. This method removes the small scale variations in OIB and BGEP data that cannot be detected from satellite, therefore making the in situ data more representative of altimeter observations.

In order to clarify these points the following sentences will be added into the article:

L390: "...between the 2 baselines. The freeboard_20_ku parameter (freeboard of the 2 baselines) is a radar freeboard, i.e the raw measurement of the freeboard without corrections (such as the snow depth).

L400: Figure 9 presents scatter comparisons with the Beaufort Gyre Exploration project (BGEP, https;//www.whoi.edu/beaufortgyre) and NSIDC Operation Ice Bridge official product (OIB, https://daacdata.apps.nsidc.org/pub/DATASETS/ICEBRIDGE/Evaluation_Products/IceBridge_Sea_Ice_Freeboard_SnowDepth_and_Thickness_QuickLook) in situ measurements. To compute OIB sea ice freeboard, we calculate the difference between the ATM mean total freeboard and the snow depth estimated from the snow radar. The freeboard radar is derived taking into account the decrease in the radar velocity in the snow pack as follows:

$$FB_{radar} = FB_{ice} - snowdepth \times (1 + 0.51 \times \rho_s)^{(-1.5)} (2)$$

with $\rho_s = 0.3$

To compare with BGEP data, we compute a CryoSat ice draft from the difference between the gridded sea ice thickness (that integrates the snow load) and ice freeboard data. Note that the ice freeboard is calculated from the radar freeboard taking into account the decrease in radar velocity in the snow pack using the formula specified in Eq 2 with the snow depth provided by the Warren99 modified climatology and the official OSI SAF sea ice type classification available at the NSIDC.

To ensure the consistency between in situ measurements and altimetric observations, all data are projected onto monthly EASE2 500x500 grids identical to the one of the altimetric product. Each in situ measurement presented in Figure 9 is the average of all data in a 12.5 x 12.5 km grid pixel size.

L409-413 – The caption is confusing. What it should say is C and D are BGEP drafts compared to drafts calculated from CS-2 freeboards.

**Reply:** the comparisons reported in Figure 9 are indeed the Baseline-C and Baseline-D freeboard data (on Y axes) versus the OIB freeboard (X axes) for figures a) and b), while the c) and d) figures report the comparison between the derived drafts from Baseline-C and Baseline-D to BGEP draft.

L559 – which water mask?

**Reply:** For Sweden: Global Lakes and Wetlands Database

Lehner, B., & Döll, P. (2004). Development and validation of a global database of lakes, reservoirs and wetlands. Journal of Hydrology, 296(1), 1–22.

For Tibet: Landsat based water mask

Jiang, L., Nielsen, K., Andersen, O. B., & Bauer-Gottwein, P. (2017). Monitoring recent lake level variations on the Tibetan Plateau using CryoSat-2 SARIn mode data. Journal of Hydrology, 544, 109–124. https://doi.org/10.1016/j.jhydrol.2016.11.024

The above references will be added to the revised version of the manuscript.

L564 – Why one meter? Where does this definition stem from? How would results change of more strict requirement (say 50 cm) would be used?

**Reply:** The one meter threshold was just chosen as a reference for comparing the two baselines. The point is to quantify the difference in valid observations between the two baselines. As suggested we could also choose a threshold of 0.5 meters as the reference. The results of this threshold are illustrated in the figure below. We will add the following sentence: "The one meter threshold is arbitrary and was simply selected to establish a common reference".

L575 – where did this offset originate from and which correction fixed it?

**Reply:** The range window extension introduced for SAR/SARIn modes in Baseline-C required that the code account for the change in reference bin position to avoid a 60 m height bias being introduced. For SARIn mode, the code was updated to fix the issue for the target surface types of ocean and continental ice, but not for other regions where the mode mask places the satellite in SARIn mode (i.e. rivers and lakes as in this case). This has been corrected in Baseline-D, removing the 60 m height bias everywhere.

References to this can be found in presentations held at Living Planet symposium of 2016 such as:

Bercher, Nicolas; Fabry, Pierre; Ambrózio, Américo; Restano, Marco; Benveniste:. Jerome: "Validation of CryoSat-2 SAR and SARin modes over rivers and lakes for the SHAPE project",

and

Borsa, Adrian: "Validation of CryoSat-2 LRM and SARIN-mode elevations over the salar de Uyuni, Bolivia"

L590 - "All kinds" however limited here to land ice, sea ice and (marginally) inland water. Rephrase.

**Reply:** done.

---

## Author Comment (AC3) · 18 Feb 2020

**Answers to Referee 3**

This article is concerned with the most recent CryoSat-2 processing and dataset version, Baseline-D, which has been operational since May 2019. The paper provides an overview of the main updates and improvements since the previous Baseline-C version, both at Level-1B and Level-2 stages. The discussed improvements at L1B stage are: Transition to the more ergonomic NetCDF file format from EEF; the eradication of anomalously negative radar freeboards at the SAR/SARIn mode boundary; the inclusion of two additional stack parameters; and inclusion of the USO correction to the window delay parameter. Improvements at the Level-2 stage are; transition to NetCDF format; the inclusion of sea ice freeboard data for SARIn mode; a new retracker for diffuse waveforms; improved surface type discrimination with the implementation of the Stack Peakiness parameter; implementation of new slope models for land ice elevation correction; and the inclusion of some additional parameters in the L2 files. The second half of the manuscript offers a series of land ice and sea ice comparisons/validations, either comparing the data with itself (e.g. during ascending and descending passes), with previous Baseline-C data, or with independent observations.

Given that the move to Baseline-D has already happened, it is important that the community understand the main changes since the previous baseline and are convinced that the new data is at least consistent if not improved. This paper offers some important findings to this end but the structure and writing need improving. I therefore recommend the paper's publication subject to the following revisions.

Main points:

1. The paper suffers some continuity issues, where sections can feel a bit disjointed and the use of terminology is not consistent throughout. In particular, the following points should be addressed:
    - I find the subsections of the land ice section (3.2) quite confusing. Section 3.2.1 'Impact of algorithm evolution on land ice products' includes different case examples over East Antarctica and Austfonna. The following sections (3.2.2 and 3.2.3) are then concerned with swath data over Antarctica and SARIn data over Austfonna. Why do these two sections not also fall under 'Impact of algorithm evolution on land ice products'? Perhaps Section 3.2.1 could be broken into a number of subsections, each with a different case example, including sections 3.2.2 and 3.2.3? Also it may flow better if the cases over Austfonna followed each other.

**Reply:** thanks to the referee for this comment. The suggestions made will be considered in the new version of the manuscript.

    - Move the first paragraph of section 3.3.3 to the beginning of section 3.3.2. Should section 3.3.3 go inside section 3.3.2 since it falls under Baseline-D freeboard assessment?

**Reply:** thanks to the referee for this relevant comment. The suggestions made will be taken into account in the new version of the manuscript to enhance readability.

    - In the abstract and elsewhere, the processors are referred to as 'Ice Baseline-C' and 'Ice Baseline-D'. However in section 3.3.4, 'IPF1C' and 'IPF1D' appear

for the first time and are used throughout this section. Please choose a name/acronym for the processors, define them in the introduction, and ensure their use is consistent throughout the manuscript.

**Reply:** thanks to the referee to have spotted this. The processor names will be homogenised through the whole manuscript, in the new version.

- As it stands, the relevance of section 3.3.5 is hard to appreciate. The section refers to re- scaling of parameters between baselines B and C, but makes no mention to the re-scaling of parameters from Baseline-C to Baseline-D, which was presumably necessary and would be of interest to the reader. Since the L2 Baseline-D processor does not use the Lee 2018 method for lead identification, more explanation is needed to tie this section in and relate it to the previous content.

**Reply:** We entirely re-write section 3.3.5 adopting stack peakiness in lead classification.

2. The article heavily cites documents (mainly ESA documents) via URLs in the text, some of which feel like necessary supplementary material to the main text (e.g. the 'CryoSat-Baseline- D-evolutions' document). Is the permanence of these URLs certain? If not - can they be put on a DOI or provided in a supplementary?

**Reply:** the cited documents are official ESA documents for which the permanence of the URLs is guaranteed.

3. Page 13, line 245 - You say that "Most of the parameters were found to show a close agreement.." I find this quite vague and expect users would want to know more on how parameters compare between each baseline. Could you include more details or a table?

**Reply:** We will change the sentence to:

"Most of the parameters were found to show agreement. The parameters showing differences are explained below an listed here: surface mask, attitude (roll, pitch, yaw), sigma0 for all LRM retrackers, slope correction (height, longitude, latitude)"

We found varying sigma0 differences for each of the retrackers as follows: Ocean: 0.6 - 0.75 dB, ICE1: 0.65 - 0.78 dB, ICE2: 0.63 - 0.77 dB

As we checked 100 different parameters for a couple of tracks, the authors preferred to not list them in a table for readability reasons.

4. The validation of LRM data over land ice depends on a comparison to REMA. Please could you provide some details about how REMA is built, e.g. what data is used in its construction, and a justification for why you chose to validate with REMA? Is there any particular reason that this area of East Antarctica was chosen?

**Reply:** REMA (Howat, 2019) was used as an independent reference elevation model. REMA is one of the most recent and accurate DEMs for Antarctica.

REMA is stated to have an absolute uncertainty of less than 1 m over most areas and was vertically registered using CryoSat and ICESat. It was constructed from optical stereo pairs from WorldView acquired between 2009 and 2017, with most collected in 2015 and 2016, over the austral summer seasons (mostly December to March). We will use mosaicked versions in two different resolutions (200m and 1km).

We selected an area on the Antarctic plateau to demonstrate the differences of the applied slope corrections. We selected the area to cover slopes from 0 to 0.25° as over 95% of the LRM mode data is acquired in low sloped area.

Only one region was selected to visualise the differences instead of showing all Antarctica.

Howat, I. M., Porter, C., Smith, B. E., Noh, M.-J., and Morin, P.: The Reference Elevation Model of Antarctica, The Cryosphere, 13, 665-674, https://doi.org/10.5194/tc-13-665-2019, 2019.

5. Why did you not validate/compare Baseline-D over land ice with an independent observation dataset like IceBridge?

**Reply:** this is a good suggestion. In this work we used the OIB dataset to validate measurements over sea ice, while for land ice we used CryoVEX campaign data. The use of OIB in land ice validations can be taken into account in future CryoSat ice data products validations.

Minor points

**Section 1:**

• Page 2, line 27 - "on 8 April 2010" -> "on the 8th April 2010"

   **Reply:** Thanks to the referee for the correction. This will be fixed in the revised version of the manuscript.

• Page 2, line 30 - Is CS2's repeat cycle not exactly 369 days with a 30-day sub cycle? Not sure what 'quasi' is referring to here.

   **Reply:** The CryoSat orbit does not exactly repeat after each cycle, as it is usually the case for ocean-oriented altimetry missions. CryoSat's ascending nodes are repeating from cycle to cycle within a few tens of meters in order to have equidistant ascending equator crossings in the reference ground track. The descending nodes are however no longer equidistant due to a residual rotation of the eccentricity vector, therefore fluctuations up to nearly 4 km can still be observed on the descending node from cycle to cycle.

• Page 2, line 50 - "with a factor 2" -> "by a factor 2"

**Reply:** Thanks to the referee for the correction. This will be fixed in the revised version of the manuscript.

- Page 3, line 53 - "CryoSat ice Baseline-D" -> "CryoSat Ice Baseline-D"

  **Reply:** Thanks to the referee for the correction. This will be fixed in the revised version of the manuscript.

- Page 4, line 71 - "affecting the Polar Regions" -> "affecting Earth's polar regions"

  **Reply:** Thanks to the referee for the correction. This will be fixed in the revised version of the manuscript.

- Page 4, line 78 - "which are contributing to global sea level rise." -> "which influence global sea level." (variations in thickness can mean thickening which does not cause sea level rise)

  **Reply:** Thanks to the referee for the correction. This will be fixed in the revised version of the manuscript.

- Page 4, line 83 - "interferometric" -> "Interferometric"

  **Reply:** Thanks to the referee for the correction. This will be fixed in the revised version of the manuscript.

- Page 4, line 88 - "The ice products are currently generated with the Ice Baseline-D processors". Please be more specific here. Since when are they generated with Baseline-D processors? covering which operational period?

  **Reply:** The transfer to operations of the new CryoSat Ice Baseline-D processors was performed on 27th May 2019 and a complete mission data reprocessing is on-going in order to provide users with homogeneous and coherent CryoSat ice products for proper data exploitation and analysis. This is specified in lines 106-109. We will move this paragraph.

- Page 5, line 106 - "sea ice and inland waters domains." -> "sea ice and inland waters."

  **Reply:** Thanks to the referee for the correction. This will be fixed in the revised version of the manuscript.

**Section 2**

• Page 7, line 139-141. Here you mention the findings of Armitage et al. 2014 but they do not relate directly to the next sentence. This sentence could be removed since you discuss the relevance of the Armitage and Davidson study to the anomalously negative freeboards in section 3.3.3

**Reply:** Thanks to the referee for the correction. This will be fixed in the revised version of the manuscript.

- Page 8, line 147. Please explain what a Doppler cell is and reference a paper e.g. Raney 1998, rather than an impermanent URL.

  **Reply:** the reference to Raney, 1998 will be added in the revised version of the manuscript replacing the URL.

- Page 8, line 148-153. Consider merging these two sentences: "At Baseline-D, two additional stack characterisation parameters (also known as Beam Behaviour Parameters) have been added to the SAR/SARIn L1B products: i) the stack peakiness (Passaro2018), which can be useful in improving sea ice discrimination, and ii) the position of the centre of the Gaussian fit......."

  **Reply:** Thanks to the referee for the correction. This will be fixed in the revised version of the manuscript.

- Page 8, line 167 - "the freeboard sea ice processing" -> "the sea ice freeboard processing"

  **Reply:** Thanks to the referee for the correction. This will be fixed in the revised version of the manuscript.

- Page 9, line 169 - "The height value..." - I don't know what height value you are referring to, could you be more specific?

  **Reply:** this will be rephrased by: "The retrieved height value…"

- Page 9, line 172 - Could you detail briefly this new retracker? Is it physical / threshold etc.

  **Reply:** the sentence will be changed to:

  "In addition, a new threshold-of-first-maximum retracker is used…"

  And after that sentence, the following text will be added:

  "Retracking is the process whereby the initial range estimate in the L1B data is corrected for the deviation in the first echo return within the waveform from the reference position."

- Page 9, line 178 - "based on the peakiness of SAR waveforms" - add "(see section 3.3.1)"

  **Reply:** Thanks to the referee for the correction. This will be fixed in the revised version of the manuscript.

- Page 9, line 187 - "..to make the correction more responsive.." - What correction? Please be more specific.

  **Reply:** this is referred to the slope correction. The paragraph will be reformulated.

- Page 10, line 194 - "in addition to the height". What height?

  **Reply:** This is a typo, the sentence between brackets will be removed.

**Section 3**

• Page 11, line 210 - "...data files was" -> "...data files were"

**Reply:** Thanks to the referee for the correction. This will be fixed in the revised version of the manuscript.

- Page 11, line 212-213 - "...geophysical corrections were checked to ensure that they were computed correctly". This is a little vague, can you say something more concrete about how they were checked?

  **Reply:** The CryoSat data products contain many data flags to which provide information and warnings about any inconsistencies present in the data products. For example, the "correction error flags" indicate whether the geo-corrections have been correctly computed during processing. These flags are checked routinely as part of operational quality control activities.

- Page 12, line 222 - "...generated using the Baseline-C are..." -> "...generated using the

  Baseline-C processors are..."

  **Reply:** Thanks to the referee for the correction. This will be fixed in the revised version of the manuscript.

- Page 12, lines 225-229 - Consider moving the sentence starting "To derive mass balance.." to after the sentence ending "..should help to reduce this uncertainty." on line 239 to aid the flow of this section.

  **Reply:** Thanks to the referee for the correction. This will be fixed in the revised version of the manuscript.

- Page 12, line 231 - "respectively mass change" - I don't know what 'respectively' means here.

  **Reply:** this is a typo. "Respectively" will be changed in "and".

- Page 12, line 234-236. Please provide a reference for this statement.

  **Reply:** the reference Gourmelen et al, 2017 will be moved at the end of the sentence.

- Page 12, line 236-237 - "However, a small attitude angle error interpreted as a mispointing error has been observed.." Observed by who? Please provide a reference.

  **Reply:** This was an issue observed in the previous Baseline-C data, now fixed in the new Baseline-D implementation.

- Page 12, line 241 - "were compute" -> "were computed"

  **Reply:** Thanks to the referee for the correction. This will be fixed in the revised version of the manuscript.

- Page 12, line 242 - "Level 2 "in depth" (L2I) product retracker" - what is this? Is there a technical note or article you could reference?

  **Reply:** Level 2 "in depth" products are a particular output product of the CryoSat Payload Data Ground Segment (PDGS).

- Page 13, line 247 - are the constant offsets on Sigma0 you list for Baseline-D minus Baseline-C or vice versa?

  **Reply:** Is referred to Baseline-C minus Baseline-D.

- Page 13, line 247-249. The sentence starting "This needs to be considered..." does not make sense, please re-phrase.

  **Reply:** thanks to the referee for this comment. The sentence will be rephrased in: "The mentioned offsets need to be considered…"

- Page 13, line 249-250 - "Furthermore, Baseline-D uses an updated surface type mask. This..." -> "A new surface type mask has been implemented in Baseline-D, significantly improving resolution in the ice shelf area...."

  **Reply:** we acknowledge the referee suggestion and we will implement it in the new version of the manuscript.

- Page 13, Figure 2. Please include a scale e.g. "Orange=Ice shelf, Blue=Ice sheet".

  **Reply:** this will be added in the figure caption.

- Page 13, Figure 2 caption: "Ronne ice shelve" -> "Ronne ice shelf"

  **Reply:** Thanks to the referee for the correction. This will be fixed in the revised version of the manuscript.

- Page 14, line 255 - "This slightly changes the LRM slope corrected elevation". What does slightly mean? 1% ? 10% ? Please quantify the change.

  **Reply:** The changes are quantified in Figure 3. We found mean differences to REMA for the slope corrected height of ICE1 retracker:

We excluded outliers which differ by more than +/-20 m.

Baseline-C:
* * *
Mean :     -0.10569497
Median :     0.10062109
Std Dev :     1.7323635
Number of points:   17050710
* * *
Baseline-D:
* * *
Mean :     -0.93375798
Median :     -0.34570312
Std Dev :     2.0853337
Number of points:   17079280
* * *
The slope model used to estimate the slope correction in the L2 product is different between both Baselines. This means that the relocated position (lat/lon) and the slope corrected elevation differs along track. Therefore, a percentage deviation cannot be specified. Figure 3 gives an idea of the spatial differences of the changes one can expect. We changed Figure 3 and show now the full LRM zone, not only a region.

[Figure]

- Page 14, line 256 - "... for a large area in East Antarctica...". Can you explain why you chose this area? Please also include a map of Antarctica or East Antarctica to show where this region is.

**Reply:** The region was chosen to cover slopes between 0 to 0.25° and to be able to distinguish differences in a figure. If needed we could add another region. We have modified Figure 3, as per previous comment.

- Page 14, Figure 3. Please say in the caption what the numbers are, i.e. "Mean REMA-CS2 difference= +0.13 ± 1.2 m" etc

    **Reply:** we acknowledge the referee suggestion and we will implement it in the new version of the manuscript.

- Page 14, line 258 - "Differences to an independent Antarctic elevation model..." -> "Differences between slope corrected elevation and an independent Antarctic elevation model..."

    **Reply:** Thanks to the referee for the correction. This will be fixed in the revised version of the manuscript.

- Page 14, line 259 - "The differences vary spatially and the overall mean..." -> The differences vary spatially and the overall mean difference (REMA minus CS2)..."

    **Reply:** Thanks to the referee for the correction. This will be fixed in the revised version of the manuscript.

- Page 14, line 265 - "...however major improvements" -> "...however offers/implies major improvements"

    **Reply:** Thanks to the referee for the correction. This will be fixed in the revised version of the manuscript.

- Page 14, line 266 - "...swath data processed for ascending and descending tracks" - For what period?

    **Reply:** this analysis covers, as for the rest of the manuscript, the test reference period (September - November 2013, February - April 2014 and April 2016 (only SARIn))

- Page 15, line 268 - "The large positive anomaly is a known.." -> "The large positive anomaly (blue area in Fig. 4) is a known.."

    **Reply:** Thanks to the referee for the correction. This will be fixed in the revised version of the manuscript.

- Page 15, line 271 - "(subpanel B) could be reduced". 1) No subpanels are labelled in the figure. 2) I don't know what you mean by "could be reduced"

    **Reply:** This is a typo. No subpanels are indeed labelled, and "could be reduced" will be changed to "is reduced", as reported in Figure 5. The text will be amended.

- Page 15, Figure 4:
    1. Please add labels "Baseline-C" and "Baseline-D" on the right-hand side and "Ascending" and "Descending" above the sub-panels.

2. Please make the labels of the colour bar larger.
3. In the caption please change "Differences to relative elevation model.." to "Differences between CryoSat elevation and reference elevation model..." or "Deviation of CryoSat elevations from reference elevation model..."

**Reply:** thank to the referee for the comment. The figure will be enhanced in the revised version of the manuscript.

- Page 16, Figure 5. "Crossovers between ascending descending.." -> "Difference in elevation between ascending and descending crossovers..."

**Reply:** Thanks to the referee for the correction. This will be fixed in the revised version of the manuscript.

- Page 17, line 291 - "a point-to-point comparison was performed". Please make clear in the text that you are not comparing Baseline-C points with Baseline-D points as this is how this reads.

**Reply**: Indeed the point-to-point comparison has been made considering swath elevations in Baseline-C and swath elevations in Baseline-D, as specified in lines 302-306.

- Page 17, line 310 - "...which is included in all comparisons." Do you mean that it is accounted for in the comparison, i.e. subtracted?

**Reply:** yes, the vertical offset is taken into account in the comparisons.

- Page 18, line 322-323 - "CryoSat operates in the new and innovative SARIn mode..." -> "CryoSat operates in SARIn mode..."

**Reply:** Thanks to the referee for the correction. This will be fixed in the revised version of the manuscript.

- Page 18, line 326 - "recommendations from the ESA project, CryoVal-LI, the 2016..." -> "recommendations from CryoVal-LI, the 2016..."

**Reply:** Thanks to the referee for the correction. This will be fixed in the revised version of the manuscript.

- Page 19, lines 333-339. "The AWI land ice processing", "NASA JPL land ice processing" and "University of Ottawa CryoSat processing", are these retrackers? or are they processors? Please tidy these distinctions up in the text.

**Reply:** those are processors which include also a dedicated retracker. It will be made clearer in the revised version of the manuscript.

- Page 20, "...before the dedicated land ice retrackers of AWI, JPL and UoO are reached." I don't know what a retracker being reached means.

> **Reply:** The sentence is misleading, it will be removed.

- Page 20, Table 1. Please change "Mean [m]", "Median [m]" and "Std. Dev. [m]" to "Mean ALS- CS2 difference [m]", "Median ALS-CS2 difference [m]", "Std. Dev. on ALS-CS2 difference [m]".

  Also state in the caption what the numbers in the brackets represent.

  > **Reply:** Thanks to the referee for the correction. This will be fixed in the revised version of the manuscript.

- Page 21, Figure 7. In caption: "CryoVex airborne laser scanning." -> "CryoVex Airborne Laser  Scanner (ALS)."

  > **Reply:** Thanks to the referee for the correction. This will be fixed in the revised version of the manuscript.

- Page 21, line 362 - "Statistics that describe the power of the stack in CryoSat were...." -> "Statistics that describe the power of the CS2 waveform stack were...."

  > **Reply:** Thanks to the referee for the correction. This will be fixed in the revised version of the manuscript.

- Page 21, line 365-366 - "This compares the maximum power registered in the Range Integrated Power (RIP) with the power obtained from the other looks". The RIP of which look? From Passaro, I understand that the Stack Peakiness "compares the power at the zero look angle with the backscatter registered in the other looks", please check your definition.

> **Reply:** The sentence of the draft is correct. The Reviewer cites only an extract of the sentence extrapolated from Passaro et al. The full sentence is "In order to compare the power at the zero look angle with the backscatter registered in the other looks, a new parameter called Stack Peakiness (SP) is defined in this study from the RIP normalised by its maximum value. The assumption is that the maximum return is (or is close to) the nadir position. This is why, as stated in Passaro et al., the application of a window (Hamming window) on the Stack data is necessary, because otherwise: "The sidelobe effects create false leading edges, influence the statistical analysis of the RIP and add backscattering of the same order of magnitude of the nadir return in the look angles closer to zero. These features mostly disappear after the application of the Hamming window, although residual signatures are visible". We agree with the Reviewer that a possible way to improve the Stack Peakiness parameter would be to consider possible inconsistencies between the look where the maximum is located and the exact nadir look. This is also why we reported in Baseline-D the Stack Peakiness as a useful complementary statistic without binding a strict classification criterion to it (exactly as it is done for the other Stack statistics).

- Page 21, line 369 - "..with the highest power (supposedly at nadir) with the looks". Again, it is the power in the nadir beam that is compared with the off-nadir looks. I understand that the RIP waveform is first normalised by its peak value- which may not be at nadir- but this sentence confuses the two steps.

  **Reply:** This has been explained in the previous reply.

- Page 22, line 372 - "The evolution of the SP over a sea-ice covered area" - I don't know what you mean by 'evolution' here. Evolution in time?

  **Reply:** Yes, is indeed the temporal evolution of the SP. This will be made clearer in the text.

- Page 22, line 382. Could you re-iterate at the end of this section that the SP parameter is implemented in lead discrimination for L2 sea ice products (as discussed in section 2.2) and mention the thresholds that are used or direct reader to where they can find the thresholds.

  **Reply:** Thanks to the referee for the correction. This will be fixed in the revised version of the manuscript.

- Page 22, line 386 - "...highlighted important over-estimations in the freeboard values of the ESA CryoSat Baseline-C products relative to in-situ". Is there a reference to support this claim? The URL is just a link to the site for CSN.

  **Reply:** This study was performed in the context of the CryoSea-Nice ESA project but this specific result was presented during the CryoSat Expert Meeting (CSEM) in November 2017 at ESA/ESRIN and documented in the Summary and Recommendations Report available at https://earth.esa.int/documents/10174/1822995/CryoSat-CSEM-Summary-and-Recommendations-Report.pdf.

  The sentence will be rephrased in the revised version of the manuscript to include this reference in the following way:

  "Previous analyses carried out by the CryoSea-Nice ESA project (https://projects.alongtrack.com/csn/) highlighted important over-estimations in the freeboard values of the ESA CryoSat Baseline-C products relative to in situ data (see the recommendation Rec.9 in [CSEM Report 2017])
  Following these conclusions, modifications have been made to develop the new ESA CryoSat Baseline-D freeboard product.
  We present here the first assessments of this updated version."

  [CSEM Raport 2017] Summary and Recommendations Report of the CryoSat-2 Expert Meeting, CSEM, 2017, ESRIN, https://earth.esa.int/documents/10174/1822995/CryoSat-CSEM-Summary-and-Recommendations-Report.pdf

- Page 22, line 390 - "...Figure 8 present the evolution between two Baselines." I think this figure is simply showing the difference rather than any evolution.

  **Reply:** yes indeed, it was a typo. The word "evolution" will be replaced with "differences".

- Page 22, line 392-393 - "...the two solutions remain consistent with each other". Could you add a comment on the larger differences in the MYI region north of Greenland?

  **Reply:** The larger differences in the MYI region north of Greenland are mainly noise at the ice margin. A comparable feature is also detected along the Russian coastline. These differences are statistically negligible but we cannot do further analysis since we do not have access to the Baseline processing chains. In order to specify this point we add the following sentence:

  *L393: The small patterns of higher differences (e.g: north of Greenland) are associated with statistically negligible noise at the ice margin zones.*

- Page 22, line 393 - "The Root Mean Square (RMS) in each..." Do you mean the Root Mean Square deviation from the average value in each pixel? i.e. the standard deviation?

  **Reply:** the sentence will be rephrased in: "In addition, the Root Mean Square (RMS) in each 20 x 20 km pixel, which represents small scale freeboard variability, is similar for the 2 Baselines (about 15 cm)."

- Page 23, line 401. Please state which OIB dataset was used - Quicklook / L2 / L4?

  **Reply:** The OIB dataset used is the NSIDC Quicklook version available at https://daacdata.apps.nsidc.org/pub/DATASETS/ICEBRIDGE/Evaluation_Products/IceBridge_Sea_Ice_Freeboard_SnowDepth_and_Thickness_QuickLook

  To specify this point we have added this URL line 401.

- Page 24, line 407 - "...with a factor..." -> "...by a factor..."

  **Reply:** Thanks to the referee for the correction. This will be fixed in the revised version of the manuscript.

- Page 24 line 414-415. I don't know what this sentence means.

**Reply:** the sentence will be rephrased in "Some additional comparisons have demonstrated that the Baseline-D freeboard solution is within the range values of recent freeboard estimations reported in Ricker et al, 2014 and Guerreiro et al, 2017.

- Page 25, line 418 - "...SARIn mode had positive impacts on sea ice freeboard". The word 'positive' here is ambiguous - do you mean positive as in good? or positive as in greater thanzero? Please clarify in the text.

  **Reply:** indeed, the sentence here is ambiguous. The sentence will be rephrased:"In addition, the improved phase difference in SARIn mode had positive impacts on the sea ice freeboard estimation from SARIn acquisition, removing negative freeboard heights at the boundary of the SARIn mode mask, as presented in the next section"

- Page 25, line 437. Why is Laxon 2003 referenced here? Laxon does not mention off-nadir leads.

  **Reply:** This is a typo. The correct reference here should be (Armitage et al., 2014):

  Armitage, T. W. K. and Davidson, M. W. J.: Using the Interferometric Capabilities of the ESA CryoSat-2 Mission to Improve the Accuracy of Sea Ice Freeboard Retrievals, IEEE Transactions on Geoscience and Remote Sensing, vol. 52, no. 1, pp. 529-536, 2014

- Page 26, line 446 - "..responsible to calibrate.." -> "..responsible for calibrating..."

  **Reply:** Thanks to the referee for the correction. This will be fixed in the revised version of the manuscript.

- Page 28, line 471 - Why is AWI listed here but none of the other groups?

  **Reply:** "AWI" word will be removed, leaving indeed only the references, for better readability.

- Page 28, line 489 - "..SAR and SIN modes of the altimeter is shown..." -> "..SAR and SIN modes of the altimeter are shown..."

  **Reply:** Thanks to the referee for the correction. This will be fixed in the revised version of the manuscript.

- Page 29, line 492 - "Its magnitude is increasing.." -> "Its magnitude increases.."

  **Reply:** Thanks to the referee for the correction. This will be fixed in the revised version of the manuscript.

- Page 29, line 505 - "We therefore speculate that the change in power scaling for SIN [...] is the reason...". Please provide further details about this change in power scaling as it's unclear what you are referring to here.

**Reply:** Thanks to the referee for the comment. This line will be rephrased in: "We therefore speculate that the change in power scaling of L1B SIN waveforms which was twice the expected waveform in Baseline-C IPF1 and now corrected in Baseline-D IPF1…"

- Page 30, Figure 11 caption: "The inner boxed indicates.." -> "The inner box indicates.."

**Reply:** Thanks to the referee for the correction. This will be fixed in the revised version of the manuscript.

- Page 32, line 551 - "...discovered that the CryoSat's altimeter.." -> "...discovered that CryoSat's altimeter.."

**Reply:** Thanks to the referee for the correction. This will be fixed in the revised version of the manuscript.

- Page 32, line 574 - "respectfully" -> "respectively"

**Reply:** Thanks to the referee for the correction. This will be fixed in the revised version of the manuscript.

- Page 35, line 597 - "with respect to previous baseline" -> "with respect to the previous baseline"

**Reply:** Thanks to the referee for the correction. This will be fixed in the revised version of the manuscript.

---

## Author Comment (AC1)

**Answers to Referee 1**

The paper is well put together and offers detailed assessment of the validity of Ice measurements at Baseline D. However, I would suggest to the authors to include more information on the specific changes/evolutions that have been implemented between the Baseline C and D processing chains. I have provided a set of minor comments and recommendations but have no significant concerns with the author's methods or results. My review is focused on the sea ice validation, since that is my area of expertise, although I have made a few minor comments elsewhere.

General comments:

1. It is important for tracking the history of each baseline to describe here what issues led to poor quality L2 data in baseline C (e.g. Section 3.3.2) and then what specific modifications were made to the retracking algorithms or processing chains that have led to vast improvements at baseline d.

**Reply:** We would like to thank the reviewer for the comment. The intention of the authors was to avoid making the text too technical, therefore reporting only the major evolutions applied to the new Ice Baseline-D. The complete list of improvements and evolutions implemented in both L1B and L2 Baseline-D processing chains is detailed in paragraphs 2.1 and 2.2, but we acknowledge the suggestion of the reviewer and we will add a summary table with the major differences between the two baselines.

Minor comments/edits: Line 40-41. Reword to explain why the 12 km is relevant.

**Reply:** The sentence will be rephrased as following: "Over sea ice, Baseline-D improves the quality of the retrieved heights inside and at the boundaries of the Synthetic Aperture Radar Interferometric (SARIn or SIN) acquisition mask, removing the negative freeboard pattern which is beneficial not only for freeboard retrieval, but for any application that exploits the phase information from SARIn Level 1B (L1B) products."

L48- 49. Are the exact same set of auxiliary measurements used for this ice draft analysis at baselines C and D?

**Reply:** The exact same set of auxiliary measurements is used to compare the Baseline C and D. In order to keep a smooth reading of the abstract we add the following sentence in section 3.3.2.

L403: The same set of auxiliary measurements is used to compare the Baseline C and D. (before the sentence: Relative to OIB....)

Fig 1. Please include product acronyms in the captions.

**Reply:** These will be added in the revised version of the manuscript.

Section 1. It would be useful here to include some introduction to the observations produced in the L2 data product. What specific measurements are provided by the ice processor at L2 for land ice, sea ice and lakes?

**Reply:** Thanks to referee 1 for this comment. The main outputs of the L2 Ice processing chain are the radar freeboard estimates, the difference in height between ice floes and adjacent waters well as ice sheet elevations, tracking changes in ice thickness. The text will be amended accordingly.

**L 140-141. How can the SARIn mode be used to reduce uncertainty?**

**Reply:** According to (Di Bella, 2018) the phase information available in the SARIn acquisition mode can be used to estimate the across-track location of leads, correct for the range overestimation and ultimately get a more precise value of the along-track SSH. The higher precision of the SSH enables, in turn, to reduce the uncertainty of the sea ice freeboard retrievals. The reference to (Di Bella, 2018) will be added at the end of the statement.

L 143. Need to explain what is meant by 'bad phase difference calibration'.

**Reply:** The phase difference calibration in Baseline-C did not consider CAL4 at the beginning of the SARIn acquisition. The statement will be rephrased and it will be added a reference to Section 3.3.3 where the issue is described together with the impact of its the in Baseline-D.

L150-153. What are these parameters for and how Cn they be used by the community?

**Reply:** The parameters are the stack peakiness and the position of the centre of the Gaussian that fits the range integrated power of the single look echoes within a stack as function of the look angle. Stack peakiness can be used to improve the sea ice discrimination. The position of the centre of the Gaussian that fits the range integrated power of the single look echoes within a stack as function of the look angle gives additional information on the shape of the Range Integrated Power, similar to the other stack characterisation parameters already present in the product.

L159. OK to refer to another study, but you need to at least include a definition here of this correction.

**Reply:** The statement will be rephrased to clarify that the mispointing angle accuracy was improved by considering a proper correction for the aberration of light when the data from Star Trackers are processed on-ground. In fact, the Star Trackers compute the satellite orientation in an inertial reference frame starting from comparison of the stars in their field of view with an on-board catalogue, therefore the aberration of light needs to be compensated for on ground to give accurate information about the satellite attitude.

L170. What is specific about the SARIn mode retracking? Specific in comparison to SAR mode?

**Reply:** The height value is still that from the SARIn mode specific retracking (phase has been used to relocate the height measurement across track), but new fields have been added to contain the sea ice processing height result (not relocated, and different retrackers for specular and diffuse waveforms), and freeboard and sea level anomalies are now computed in SARIn mode (previously SAR mode only).

L172-173. Define retracking before this discussion. You also need to include details of this retracker and how it is implemented.

**Reply:** the sentence will be changed to:

"In addition, a new threshold-of-first-maximum retracker is used..."

And after that sentence, the following text will be added:

"Retracking is the process whereby the initial range estimate in the L1B data is corrected for the deviation in the first echo return within the waveform from the reference position."

L176. 'Records' is quite ambiguous. Returns?

Reply: thanks for the comment, replaced with "waveform".

L214-215. This was an issue with baseline c data, or just an issue with the selected TDS for baseline d?

**Reply:** this was an issue with the Baseline-C processing chain in general and not specific to the particular TDS used in this study. The issue has now been fixed in Baseline-D.

L238-239. Clarify whether the angular correction is implemented by the data provider for baseline d L1B products? Can you explain in a little detail here the source of the angular error and its spatiotemporal dependence?

**Reply:** In Baseline-D a new Star Tracker Processor was developed to create files containing the most appropriate Star Tracker data. In addition, new fields were added to the L1B products to include the antenna bench angles (roll, pitch and yaw) and the sign conventions of these fields were updated.

L247. What are these retrackers? What are their differences? It would be extremely useful generally for the altimetry ice community if the authors could provide a table here with details of all the retrackers implemented for each surface type and sensing mode.

**Reply:** The details about the implemented Baseline-D retrackers are given in the Product Handbook document available at:

https://earth.esa.int/documents/10174/125272/CryoSat-Baseline-D-Product-Handbook

This reference will be added in the revised version of the manuscript.

L249. Citations?

**Reply:** the following citations will be added to the revised version of the manuscript.

Simonsen, S. and Sørensen, L.: Implications of changing scattering properties on Greenland ice sheet volume change from Cryosat-2 altimetry, Remote Sens. Environ., 190, 207–216, https://doi.org/10.1016/j.rse.2016.12.012, 2017.

Schröder, L., Horwath, M., Dietrich, R., Helm, V., van den Broeke, M. R., and Ligtenberg, S. R. M.: Four decades of Antarctic surface elevation changes from multi-mission satellite altimetry, The Cryosphere, 13, 427–449, https://doi.org/10.5194/tc-13-427-2019, 2019.

L250. Up- dated surface mask derived from what? By whom? Fig 3. Include an inset map of the location.

**Reply:** The Level 2 products contain a flag word, provided at 1 Hz resolution, to classify the surface type at nadir. This classification is derived using a four-state surface identification grid, computed from a static Digital Terrain Model 2000 (DTM2000) file provided by an auxiliary file to the processing chain. We will add an inset map in figure 3.

L327-328. Explain why.

**Reply:** The sentence:**

"... projects. Based on recommendations from the ESA project, CryoVal-LI, the 2016 CryoVEx airborne campaign (Skourup et al. 2018) revised the traditional satellite under-flights to fly parallel lines with spacing of 1 or 2 km next to the CryoSat nadir ground tracks."

Will be revised to

Traditional airborne validation campaigns for satellite radar altimetry have targeted satellite under-flights as close to the satellite nadir as possible. This approach is favourable when surveying a flat surface, however, a sloping surface will induce an off-nadir pointing of the radar returns, and the number of coinciding observations will be limited. The ESA project CryoVal-LI quantified this off-nadir pointing based on CryoSat SARIn L2 data and based on the project recommendations, the 2016 CryoVEx airborne campaign (Skourup et al. 2018) revised the traditional satellite under-flights to fly parallel lines with a spacing of 1 or 2 km next to the CryoSat nadir ground tracks. Figure 7 shows the Austfonna flight path, which is optimised to ensure as many coinciding observations between CryoSat and airborne surveys, within the possible range of the aircraft.

L349-350. Add explanation on the latest ESA baseline d retracking algorithm and processing chain. Does it follow one of the other group's processing chains? Are the retracking solutions from other group's algorithms available in the baseline d L2 ice processor data product?

**Reply:** The details about the implemented Baseline-D retrackers are given in the Product Handbook document available at:

https://earth.esa.int/documents/10174/125272/CryoSat-Baseline-D-Product-Handbook

This reference will be added in the revised version of the manuscript.

In Baseline-D Ice L2 products only the retrackers described in the above document have been used.

L375-376. Clarify.

**Reply:** It is unclear what the reviewer refers to, since Lines 375-376 are in the middle of two sentences. If the Reviewer refers to "The Lead areas...minimum" (Lines 376-377), then the Clarification is already provided in the subsequent sentence (377-380).

If the reviewer refers to "...but the lead returns also influence the measurements nearby", this can be reformulated as "...is easily identifiable; the measurements close to the peak are characterised by a decay SP, which is still higher than the value found in the absence of a lead, since the latter can be the dominant return in the waveform up to about 1.5 km away from the sub-satellite point (Armitage et al., 2014) "

L385.The hyperlink doesn't seem to work.

**Reply:** The sentence will be rephrased in the revised version of the manuscript to include updated reference in the following way:

"Previous analyses carried out by the CryoSea-Nice ESA project (https://projects.alongtrack.com/csn/)- highlighted important over-estimations in the freeboard values of the ESA CryoSat Baseline-C products relative to in-situ data (see the recommendation Rec.9 in [CSEM Report 2017])

Following these conclusions, modifications have been made to develop the new ESA CryoSat Baseline-D freeboard product. We present here the first assessments of this updated version."

[CSEM Raport 2017] Summary and Recommendations Report of the CryoSat-2 Expert Meeting, CSEM, 2017, ESRIN, https://earth.esa.int/documents/10174/1822995/CryoSat- CSEM-Summary-and-Recommendations-Report.pdf

L394. Is this correct? I expect this rms measure is a convolution of the noise with valid signal at the sub grid-cell level. A better estimate for the noise distribution would be obtained from along-track rms of height observations over smooth level ice. Fig 8. Very difficult to see the difference map. Can you enlarge the points and ensure the color scale is cantered so that white = zero. Almost impossible to visualize the positive anomalies here.

**Reply:** We do agree with the comment that "real" RMS should be calculated along track. This is actually the procedure we use to estimate freeboard uncertainties in the products. Here we wanted to insist on the fact that the Baseline-D improvement is more a bias correction than a decreasing of noise in the product. We agree that this sentence is confusing, not necessary and not entirely true. Then we have reformulated the sentence as it follows:

L393: In addition, the Root Mean Square (RMS) in each 20 x 20 km2 pixel, referring for a small scale freeboard variability, is similar for the 2 Baselines (about 15 cm).

For a better visibility, all figures have been replotted (maps for figure 8 are given at the end of

the document).

However, a colour scale centred on zero does not provide much information (see the figure below).

mean: 0.11 m stdev: 0.03 m min: -0.60 m max: 1.03 m

mean: 0.11 m stdev: 0.03 m min: -0.60 m max: 1.03 m

---

## Author Response (AR1)

**1 CryoSat Ice Baseline-D Validation and Evolutions**

[revised manuscript text omitted]

used for retracking diffuse waveform
and for all waveforms in non-polar re |
| 230  | peakiness, and standard deviation of the stack of waveforms as metrics, in addition to peakiness  |              | the CryoSat Design Summary Docum
https://earth.esa.int/documents/10174
Design-Summary-Document).               |
| 231  | of the stack (see section 3.3.1), This method, improves the capability of the algorithm to reject |              | Deleted: records                                                                                                     |
|      |                                                                                                   | $\mathbb{N}$ | Deleted:                                                                                                             |
| 232  | waveforms contaminated by off-nadir specular reflections (as described in                         |              | Formatted: Font: Times New Roman,                                                                                    |
| 233  | https://earth.esa.int/documents/10174/125272/CryoSat-L2-Design-Summary-Document).                 |              | Deleted: the peakiness of the stack of waveforms.                                                             |
| 234  | Some tuning of the thresholds for the other metrics has also been performed based on analysis     |              | Formatted: Font: Times New Roman,                                                                                    |
|      | poine tailing of the thresholds for the other metrics has also been performed, based on analysis  |              | Deleted:                                                                                                             |
| 235  | of the test datasets. For the land ice domain, new slope models have been generated, using the    |              |                                                                                                                      |
| 236  | Digital Elevation Models (DEMs) of Antarctica and Greenland described in Helm et al. (2014).      |              |                                                                                                                      |
| 237  | These models were created with more recently acquired data and therefore better represent the     |              |                                                                                                                      |
| 238  | slope of the surface during the period of the CryoSat mission. The DEMs were sampled at high      |              |                                                                                                                      |
| 239  | resolution to derive the surface slope correction, Lastly, several improvements have been made    |              | Deleted: to make the correction more in slope                                                                 |
| 240  | to the contents of the L2 products. The surface type mask model used to discriminate different    |              |                                                                                                                      |
| 241  | types of targets, has been updated (as described in the Baseline-D product handbook available     |              |                                                                                                                      |
| 242  | at https://earth.esa.int/documents/10174/125272/CryoSat-Baseline-D-Product-Handbook ).     |              |                                                                                                                      |
| 243  | Variables have been added to the netCDF to explicitly cross-reference the 1 Hz and 20 Hz data.    |              |                                                                                                                      |
| 244  | Finally, the retracker-corrected range to the surface has been added to the productThe table      |              | Deleted: (in addition to the height).                                                                                |
| 245  | below summarizes the major differences between the Baseline-D and the Baseline-C.                 |              | Deleted:                                                                                                             |
| 0.46 |                                                                                                   |              |                                                                                                                      |
| 240  |                                                                                                   |              |                                                                                                                      |

Deleted: The height value is still that from the SARIn mode specific retracking, but new fields have been added to contain the sea ice processing height result, and freeboard and sea level anomalies are now computed in SARIn mode (previously SAR mode only). In addition, a new retracker is used for retracking diffuse waveforms from sea ice regions, and for all waveforms in non-polar regions (more details in the CryoSat Design Summary Document available at https://earth.esa.int/documents/10174/125272/CryoSat-L2-Design-Summary-Document).

**Deleted:**

| eleted: to make the correction more responsive to chang | es |
|---------------------------------------------------------|----|
| a slone                                                 |    |

247

| Table 1 Major Baseline-D evolutions    |                                          |   |  |  |  |  |  |
|----------------------------------------|------------------------------------------|---|--|--|--|--|--|
| L1b                                    | L2                                |   |  |  |  |  |  |
| NetCDF Format                          | NetCDF Format                            | • |  |  |  |  |  |
| Phase Difference Calibration           | SARIn Mode height bias corrected         | • |  |  |  |  |  |
| SARIn Scaling factor now applied       | SARIn Mode sea ice processing            | • |  |  |  |  |  |
| Stack peakiness and position of center | Sea Ice retracker for retracking diffuse | • |  |  |  |  |  |
| of Gaussian parameters added           | waveforms from sea-ice regions, and for  |   |  |  |  |  |  |
|                                        | all waveforms in non-polar regions.      |   |  |  |  |  |  |
| USO Correction included at L1b         | Sea-Ice Discrimination improved by       | • |  |  |  |  |  |
|                                        | using the new Stack Peakiness parameter  | 1 |  |  |  |  |  |
| Mispointing angles accuracy            | Improved Slope Model                     | • |  |  |  |  |  |
| increased by considering the           |                                          |   |  |  |  |  |  |
| aberration correction                  |                                          |   |  |  |  |  |  |

Formatted: Font: (Default) Times New Roman, 9 pt, Bold, Not Italic, Font color: Text 1, English (UK) Formatted: Caption, Centered, Keep with next Formatted: Font: 9 pt, Bold, Font color: Text 1 Formatted: Font: Bold Formatted: Centered Formatted Table Formatted: Font: Not Bold Formatted: Justified Formatted: Justified Formatted: Font: Not Bold Formatted: Justified Formatted: Justified Formatted: Justified Formatted: Font: Not Bold Formatted: Font: Not Bold

**268 3 CryoSat Ice Baseline-D Validation of Test Dataset Results**

| 269 | 3.1 Data Quality: Ice Baseline-D Test Data Verification by IDEAS+                                |        |
|-----|--------------------------------------------------------------------------------------------------|--------|
| 270 | All CryoSat data products are routinely monitored for quality control by the ESA/ESRIN           |        |
| 271 | Sensor Performance, Products and Algorithms (SPPA) office with the support of the Instrument     |        |
| 272 | Data quality Evaluation and Analysis Service (IDEAS+). In preparation for the Ice Baseline-      |        |
| 273 | D, IDEAS+ performed Quality Control (QC) checks on test data generated with the new Ice          |        |
| 274 | Baseline-D processors (IPF1 vN1.0 & IPF2 vN1.0). For testing and validation purposes a 6-        |        |
| 275 | month TDS was generated at ESA on a dedicated processing environment for two periods:            |        |
| 276 | September - November 2013; February - April 2014. IDEAS+ performed QC of a 10-day                |        |
| 277 | sample of L1B and L2 data, to assess data quality and check for major anomalies. Following       |        |
| 278 | this QC checks, this 6-month TDS was made available to the CryoSat QWG for more detailed         |        |
| 279 | scientific analysis.                                                                             |        |
| 280 | The content of the product header files (.HDR) was checked to confirm that all Data Set          |        |
| 281 | Descriptors (DSDs) were present and correct and all header fields were correctly filled.         |        |
| 282 | Similarly, the global attributes section of the netCDF has been checked to ensure data files     |        |
| 283 | were consistent and complete. The CryoSat data products contain many data flags to which         | (      |
| 284 | provide information and warnings about any inconsistencies present in the data products. These   |        |
| 285 | flags have been checked for any unexpected values, that may indicate processing anomalies,       | $\leq$ |
| 286 | and all external geophysical corrections were checked to ensure that they were computed          | -(     |
| 287 | correctly. Some minor unexpected changes to the configuration of particular flags was            |        |
| 288 | observed as well as the incorrect scaling of the altimeter wind speed values. These minor issues |        |
| 289 | have been resolved in the final Baseline-D release, which has been implemented into              |        |
| 290 | operations.                                                                                      | 1      |
| 291 | τ                                                                                                |        |
|     |                                                                                                  |        |

| Deleted: 1 | The test Baseline-D products were also checked for |
|------------|----------------------------------------------------|
| Deleted: f | lags                                               |

**297 3.2 Land Ice**

**298 **3.2.1** Impact of algorithm evolution on land ice products**

299 CryoSat L1B and L2 products generated using the Baseline-C processors are the primary input 300 to obtain elevation change time series of the large ice sheets. As those time series are the 301 primary data set to obtain ice sheet wide mass balance and therefore the contribution to sea 302 level change, a consistent high quality CryoSat L1B/2 product is essential. To derive mass 303 balance estimates the Alfred Wegener Institute (AWI) processing chain was used, introduced 304 by Helm et. al. 2014, including TFMRA (Threshold First-Maximum Retracker Algorithm) re-305 tracking and the refined slope correction (Roemer, et. al., 2007) for LRM mode as well as an 306 interferometric processing using phase and coherence for the SARIn mode L1B data products. 307 In addition, several other groups rely on high quality L1B and L2 data products to generate 308 time series of elevation and mass change (e.g. Nilsson et al., 2015; Simonsen et al, 2017; 309 McMillan et al., 2014; Schroeder et al, 2019). Next to the conventional along track processing, 310 the swath mode has been developed and explored by several groups (Gray et al., 2013; 311 Gourmelen et al., 2017). It has been demonstrated that swath products can be used to estimate 312 basal melt rates of ice shelves or high-resolution elevation change time series within the steep 313 margins of the Greenland ice sheet or Arctic Ice Caps (Gourmelen et al., 2017). However, a 314 small attitude angle error interpreted as a mispointing error has been observed using Baseline-315 C products, which is critical for the accuracy of the derived swath mode products. Bouffard et al., 2018b presented an attitude correction to be applied to Baseline-C products, which should 316 317 help to reduce this uncertainty. This has been implemented In Baseline-D, where a new Star 318 Tracker Processor was developed to create files containing the most appropriate Star Tracker 319 data. In addition, new fields were added to the L1B products to include the antenna bench 320 angles (roll, pitch and yaw) and the sign conventions of these fields were updated. To estimate 321 the impact of the algorithm evolution of the CryoSat Ice Processor to Baseline-D on land ice

| 324 | data records, l  | L2 type pro          | ducts for l      | Baseline-C an               | d Baseline-I         | O were o        | compute   | e d usin | g the AWI          |                        |                                                                                                                                                 |
|-----|------------------|----------------------|------------------|-----------------------------|----------------------|-----------------|-----------|-----------------|--------------------|------------------------|-------------------------------------------------------------------------------------------------------------------------------------------------|
| 325 | processing       | chain.               | In               | addition,                   | Level                | 2               | "In-o     | lepth"          | (L2I_              |                        |                                                                                                                                                 |
| 326 | https://earth.es | sa.int/docun         | nents/1017       | 74/125272/Cr                | yoSat-Basel          | ine-D-P         | roduct-   | Handbo          | ook )       |                        |                                                                                                                                                 |
| 327 | product retrac   | ker and slo          | pe correc        | tions were in               | nplemented           | in the ii       | ndividu   | al data         | sets to be         |                        | Deleted:                                                                                                                                        |
| 328 | compared. In     | a first instar       | ice single       | tracks crossir              | g the Antarc         | ctic ice s      | heet we   | re com          | pared on a         |                        |                                                                                                                                                 |
| 329 | point to point   | basis for all        | l of the ind     | dividual parai              | neters includ        | led in th       | e L1B     | and L2          | I products.        |                        |                                                                                                                                                 |
| 330 | Most of the pa   | arameters w          | vere found       | l to show clos              | se agreemen          | t, howe         | ver a co  | nstant          | offset was         |                        |                                                                                                                                                 |
| 331 | found for        | sigma0               | for al           | l of the                    | impleme              | nted            | LRM       | L2              | retrackers         |                        |                                                                                                                                                 |
| 332 | (https://earth.e | esa.int/docu         | ments/101        | 74/125272/C          | ryoSat-Base          | line-D-I        | Product   | Handb           | ook): 0.6   |                        | Deleted: :                                                                                                                                      |
| 333 | dB, 0.63 dB, 0   | ).65 dB for          | Ocean, Ice       | e1, Ice2 retrac             | ker respectiv        | vely. Th        | e menti   | oned o          | ffsets need        |                        | Formatted: Font: (Default) Times New Roman, 12 pt, English (US)                                                                          |
| 334 | to be considered | ed, as long a        | s both Bas       | selines are use             | d in combina         | ation to e      | estimate  | elevat          | ion change         |                        | Formatted: Font: (Default) Times New Roman, 12 pt, English (US)                                                                                 |
| 335 | time series, a   | as some gr           | oups inco        | orporate a si               | gma0 correl          | lated co        | orrection | (Sim     | onsen and          |                        | Formatted: Default Paragraph Font, Font: (Default) Times
New Roman, 12 pt, English (US)                                                      |
| 336 | Sørensen, 201    | 7 and Schrö          | öder et al.,     | 2019) . A nev | v surface typ        | e mask          | has bee   | n imple         | emented in         | $\setminus$            | Formatted: Font: (Default) Times New Roman, 12 pt, Font color: Auto, English (US)                                                               |
| 225 |                  |                      |                  | a                           |                      | 10              |           |                 |                    | $\langle \rangle$      | Deleted: This needs to be considered                                                                                                            |
| 337 | Baseline-D, si   | gnificantly          | improvin  | g resolution 1              | n the ice she        | elf area        | as show   | 'n in Fi | igure 2 for        |                        | Formatted: Font: (Default) Times New Roman, 12 pt, Font color: Auto, English (US)                                                               |
| 338 | the Filchner-F   | Ronne ice s          | helf. The | Level 2 proc                | lucts contair        | 1 a flag | word,     | provide         | ed at 1 Hz         |                        | Formatted: Font: (Default) Times New Roman, 12 pt, English (US), Not Raised by / Lowered by                                                     |
| 339 | resolution, to   | classify the         | surface ty       | pe at nadir. T              | his classifica       | ation is c      | derived   | using a  | a four-state       | $\left  \right\rangle$ | Deleted: Furthermore, Baseline-D uses an updated surface type mask                                                                              |
| 340 | surface identif  | fication grid        | , compute | d from a statio             | e Digital Ter | rain Mo         | del 200   | ) (DTN   | 42000) file |                        | Formatted: Font: (Default) Times New Roman, 12 pt, Font color: Auto, English (US)                                                               |
| 341 | provided by a    | n auxiliary f | file to the      | processing ch               | ain                  |                 |           |                 |                    |                        | Deleted: This has significantly improved in the ice shelf area around Antarctica, as shown in Figure 2 for the Filchner Ronne ice shelf. |